# A one-dimensional temperature and age modeling study for selecting the drill site of the oldest ice core near Dome Fuji, Antarctica

Takashi Obase[1], Ayako Abe-Ouchi[1,2], Fuyuki Saito[3], Shun Tsutaki[2,4], Shuji Fujita[2,4], Kenji Kawamura[2,3,4] and Hideaki Motoyama[2,4]

[1] Atmosphere and Ocean Research Institute, The University of Tokyo, Kashiwa, Japan
[2] National Institute of Polar Research, Research Organization of Information and Systems, Tachikawa, Japan
[3] Japan Agency for Marine-Earth Science and Technology (JAMSTEC), Yokosuka, Japan
[4] The Graduate University for Advanced Studies, SOKENDAI, Tachikawa, Japan

*Correspondence to*: Takashi Obase (obase@aori.u-tokyo.ac.jp)

**Abstract.** The recovery of a new Antarctic ice core spanning the past ~1.5 million years will advance our understanding of climate system dynamics during the Quaternary. Recently, glaciological field surveys have been conducted to select the most suitable core location near Dome Fuji (DF), Antarctica. Specifically, ground-based radar-echo soundings have been used to acquire highly detailed images of bedrock topography and internal ice layers. In this study, we use a one-dimensional (1-D) ice flow model to compute the temporal evolutions of age and temperature, in which the ice flow is linked with not only transient climate forcing associated with past glacial–interglacial cycles, but also transient basal melting diagnosed along the evolving temperature profile. We investigated the influence of ice thickness, accumulation rate, and geothermal heat flux on the age and temperature profiles. The model was constrained by the observed temperature and age profiles reconstructed from DF ice-core analysis. The results of sensitivity experiments indicate that ice thickness is the most crucial parameter influencing the computed age of the ice because it is critical to the history of basal temperature and basal melting, which can eliminate old ice. The 1-D model was applied to a 54 km-long transect in the vicinity of DF and compared with radargram data. We found that the basal age of the ice is mostly controlled by the local ice thickness, demonstrating the importance of high spatial resolution surveys of bedrock topography for selecting ice-core drilling sites.

## 1. Introduction

Earth's climate system experienced glacial–interglacial cycles during the Quaternary, associated with the waxing and waning of continental ice sheets and climate system feedbacks (e.g., Shakun et al., 2015). Ice cores from the Antarctic ice sheet have provided fruitful information on past climate system changes because they can provide continuous reconstructions of atmospheric compositions and temperature up to ~800 thousand years before the present (ka BP) (Jouzel et al., 2007; Kawamura et al., 2017). Such reconstructions have contributed to our understanding of the climate system dynamics of glacial–interglacial cycles (e.g., Abe-Ouchi et al., 2013; Obase et al., 2021). Meanwhile, a stacked sequence of marine sediments (Lisiecki and Raymo, 2005) indicates that the periodicity of glacial–interglacial cycles changed from 40 to 100 ka at the middle Pleistocene transition (MPT, approximately 800–1250 ka BP, Paillard, 2001; Clark et al., 2006). However, continuous ice core records that cover the MPT are still lacking, leading to a limited understanding of the mechanisms of this climate event. To help remedy this issue, the International Partnership for Ice Core Sciences (IPICS) has identified the quest for an "oldest ice core" as a critical scientific challenge. In this article, we define the term "old ice" as a continuous ice core with a basal age reaching 1.5 million years (Ma) BP, as defined in a IPICS community paper (Fischer et al., 2013).

In recent years, international efforts have been made to find plausible sites to obtain old ice in several locations in the interior of the Antarctic continent. In particular, in EPICA (European Project for Ice Coring in Antarctica) Dome C (EDC), glaciological surveys and ice-flow modeling

studies have been used to select the location of suitable sites (Parrenin et al., 2017; Young et al., 2017; Passalacqua et al., 2018; Lilien et al., 2021). The present article focuses on Dome Fuji (DF), Antarctica, which is located at 77.31° S, 39.70° E, with a surface elevation of 3810 m above sea level, and ice thickness of 3028 m. The most recent ice core at DF was obtained between 2003 and 2006 (Motoyama et al., 2021). The ice age at the bottom of this core was approximately 720 ka BP based on Antarctic ice core chronology 2012 (AICC2012) (Kawamura et al., 2017; Uemura et al., 2018). The temperature of the ice was at the pressure-melting point near the bedrock (Motoyama et al., 2021). Recently, field surveys have been conducted to collect bedrock elevation data near DF using ground and airborne radar surveys. On the basis of surveys performed by Japanese Antarctic Research Expeditions (JARE) between the late 1980s and 2008, the results of which are included in BEDMAP 2 and 3 datasets (Fretwell et al., 2013; Frémand et al., 2022), the typical ice thickness around DF is approximately 2000–3200 m (Fig. 1). Subsequently, the 54th JARE (2012–2013 Antarctic summer) conducted ground-based radar surveys in areas where subglacial mountains were detected in the area south of DF (data compiled in Tsutaki et al., 2022). More recently, the Alfred Wegener Institute (AWI) in Germany conducted airborne radar surveys covering the DF area (Karlsson et al., 2018). On the basis of these data, the 59th and 60th JARE (2017–2018 and 2018–2019 Antarctic summers) conducted ground-based radar surveys to investigate the internal reflection horizons (internal layers) of ice sheets over a distance of ~ 5650 km (Tsutaki et al. 2022), covering the DF and NDF sites (the latter located at 77.8° S, 39.05° E, south of DF) (Rodrigez-Morales et al., 2020).

To select suitable ice-core drilling sites, the conditions that are required to preserve old ice using constraints from glaciological and climatological data should be investigated. Previous ice-flow modeling studies have examined the requirements to preserve old ice using both three-dimensional (3-D) and one-dimensional (1-D) models. Pattyn (2010) used a 3-D ice sheet model under present-day constant climate forcing, and suggested the importance of minimal horizontal flow and low geothermal heat flux (GHF) to preserve old ice near the base of ice sheets. Other studies have used 3-D models to represent 3-D ice-flow fields and ice age for the relatively small area near Antarctic Domes (Huybrechts et al., 2007; Seddik et al., 2011; Sun et al., 2014; Passalacqua et al., 2018; Zhao et al., 2018). These studies estimated the age distribution of the ice expected from 3-D ice flow fields under a constant present-day climate. More recent studies used glacial–interglacial cycle forcing (Sutter et al., 2019, 2021) and discussed how the past variation of the Antarctic ice sheet affects age distributions of ice.

One-dimensional vertical ice-flow models have been used to estimate the vertical profiles of age and temperature near Antarctic domes, where horizontal flow is relatively minor. Horizontal surface velocity in the vicinity of DF and NDF is $< 2$ m a$^{-1}$, and it has minor spatial variations, evidenced by satellite-based measurements (Rignot et al., 2011, 2017; Mouginot et al., 2012). Such 1-D models perform well in long-term forward simulations over glacial cycles and are able to conduct many simulations with different parameters. In particular, Fischer et al. (2013) investigated the influence of a wide range of parameters, including ice thickness, accumulation, and GHF on the basal age of ice. Their key finding was that melting at the base reduces the likelihood of old ice, and a lower ice thickness than that at previous ice core sites is a required condition to avoid basal melting. Furthermore, a lower accumulation rate generally contributes to increasing the age of the ice at a certain height from the bedrock but increases the chance of basal melting, owing to the reduced vertical advection of cold ice. Other studies used an equivalent 1-D ice-flow model, investigated the necessary conditions to keep the ice base frozen (Van Liefferinge and Pattyn, 2013; Van Liefferinge et al., 2018), and examined the observed basal conditions of the ice (Passalacqua et al., 2017). Parrenin et al. (2017) estimated ice-flow parameters and basal melting rate using internal layers of the ice near EDC and proposed candidate sites for old ice. The reasonable resolution of ice core containing climate signals which can be analyzed with current methods is important. Particularly, Saito et al. (2020) presented a numerical scheme of ice advection calculation for an improved

representation of annual layer thickness of the ice, and conducted numerical simulations using idealized glacial cycle forcings.

Simplified factors in previous modeling studies were the time-dependent climate forcing and temperature profile, which are critical to basal ice melting. In particular, the basal temperature of the ice sheet shows a minimum during interglacials because it takes a long time to advect and diffuse surface temperature changes to the base of the ice sheet (Saito and Abe-Ouchi, 2004; Van Liefferinge et al., 2018). In this context, the model used in Parrenin et al. (2007) assumed that basal melting rates were constant over time, and Fischer et al. (2013) used pseudo steady-state assumption, i.e., a constant climate forcing. Parrenin et al. (2017) assumed that the temporal variations in basal melting rates are the same as accumulation rates. Some studies (Van Liefferinge and Pattyn, 2013; Passalacqua et al., 2017; Van Liefferinge et al., 2018) have investigated ice temperature using realistic climate forcing but did not investigate the resultant impact on the age of the ice. Similarly, Hondoh et al. (2002) and Talalay et al. (2021) estimated GHF at DF and other Antarctic domes based on observed vertical temperature profiles, but the observed age–depth profiles were not used as constraints. The ice thickness at Antarctic domes also changes with time, and can be up to 150 m thinner during glacial periods when surface mass balance (SMB) is reduced (Saito and Abe-Ouchi, 2010).

Despite the close link between the temperature and age of ice owing to basal melting, the coupled simulations of thermodynamics and age of ice were not represented under transient climate forcing in previous modeling studies of old ice. In this study, we use a 1-D ice-flow model, which simultaneously computes the evolution of ice temperature and age, and the model is forced by past climate history. The remainder of the article is organized as follows: Section 2 describes the 1-D model used in this study. In Sect. 3, we apply this model to DF and conduct systematic sensitivity experiments to calibrate GHF and a tuning parameter of the vertical profile of ice velocity by comparing simulated age and temperature profiles with observations. We also use parameters at EDC to examine whether the model can simulate temperature and age profiles under different glaciological conditions. In Sect. 4, using the results of the tuned vertical velocity parameters, we investigate the influences of ice thickness, SMB, and GHF on the basal temperature and age. In Sect. 5, we apply the 1-D model to the DF–NDF transect (over a distance of ~ 50 km ) and compare the results with the internal layers of the ice.

## 2. Method
## 2.1. Model description

We used a 1-D ice-flow model, IcIES-2 (Saito et al., 2020). This model computes the temporal evolutions of the age and temperature profiles of ice columns.

The evolution of the age of the ice is computed using the vertical advection equation,

$$\frac{\partial A}{\partial t} = -w \frac{\partial A}{\partial z} + 1, \quad (1)$$

where $A$ is the age of the ice, defined as the duration since deposition, and $w$ is the vertical velocity of the ice (a positive value indicates upward velocity). Here, $\zeta$ is a normalized coordinate defined as $\zeta = \frac{z}{H}$, where $z$ is the height above bedrock, and $H$ is the ice thickness (thus $\zeta = 1$ and 0 correspond to the ice surface and base, respectively). The first and second terms on the right-hand side of Equation (1) represent the vertical advection and aging owing to time-lapse, respectively.

The vertical velocity of the ice is assumed to be represented as:

$$w(\zeta) = -\left[\left(M_s + M_b - \frac{\partial H}{\partial t}\right)\omega(\zeta) - M_b\right], \quad (2)$$

where the terms $M_s$ and $M_b$ represent surface (positive indicates ice gain) and basal (negative indicates ice melt) mass balance caused by accumulation and basal melting, respectively, and $\frac{\partial H}{\partial t}$ is the change in ice thickness over time. The normalized vertical velocity profile, $\omega$, is given as a function of the normalized coordinate derived from Parrenin and Hindmarsh (2007), and Llibtoury

(1979):

$$\omega(\zeta) = 1 - \frac{p+2}{p+1}(1-\zeta) + \frac{1}{p+1}(1-\zeta)^{p+2}, \text{(3)}$$

where $\omega$ is 1 at the surface and 0 at the base. Hence, in the case of steady state, $\frac{\partial H}{\partial t} = 0$, the vertical velocity of the ice at the surface and base equates to $-M_s$ and $M_b$, respectively. The shape of $\omega$ with different $p$ parameters is demonstrated in Fig. 2, indicating that a larger $p$-value yields a larger downward ice velocity. Compared with Fischer et al. (2013), who used a different formulation of the vertical velocity profile with an $m$ parameter (similar role as $p$ of this study) of $m = 0.5$ (Fig. 2 dashed lines), $p = 3$ from Equation (3) gives a different vertical temperature profile, with a smaller vertical velocity, particularly near the base of the ice.

The temperature of the ice is computed using the following vertical advection and diffusion equation:

$$\frac{\partial T}{\partial t} = -w \frac{\partial T}{\partial z} + \frac{1}{\rho_I c_P} \frac{\partial}{\partial z}\left(\kappa \frac{\partial T}{\partial z}\right), \text{(4)}$$

where $T$ is the temperature of the ice [K], $\kappa$ is the thermal conductivity, $\rho_I$ is the ice density, and $c_p$ is the heat capacity of the ice. The density of ice is set as a constant (910 kg m$^{-2}$), i.e., we ignore the effects of lower density in the firn column. The strain heating term is neglected in the present study, given that deformation of the ice would be minor near Antarctic domes because of very low horizontal shear. The thermal conductivity and specific heat capacity of the ice are functions of temperature (Greve and Blatter, 2009, following Ritz, 1987):

$$\kappa = 9.828 e^{-0.0057T} \text{ W m}^{-1}\text{K}^{-1}, \text{(5)}$$

$$c_p = (146.3 + 7.253T) \text{ J kg}^{-1}\text{K}^{-1}, \text{(6)}$$

Boundary conditions at the surface and base of the ice are required to close the equations. At the ice surface, the age is set as 0, assuming no surface melt, and the temperature is set to the surface temperature at the given time. The basal boundary conditions for temperature depend on the basal condition:

$$\frac{\partial T}{\partial z}\big|_b = -\frac{G}{\kappa} \text{ if no melting, (7)}$$

$$T_b = T_{pm} \text{ if melting, (8)}$$

where $G$ is the geothermal heat flux (GHF) at the ice–bedrock boundary, and $T_{pm}$ is the pressure-melting point of the ice, which is given as a function of depth using a Clausius–Clapeyron gradient (8.7 x 10$^{-4}$ K m$^{-1}$). The basal melting rate at the ice–bedrock interface is determined by the conservation of heat:

$$M_b \rho_I L = G - \kappa \frac{\partial T}{\partial z}, \text{(9)}$$

where $L$ is the latent heat of the ice (335 kJ kg$^{-1}$), and $\frac{\partial T}{\partial z}\big|_b$ is the temperature gradient at the ice–bedrock interface. This model assumes basal melting only occurs at ice–bedrock interfaces, and the temperature gradient at the ice–bedrock interface is calculated using a one-sided difference discretization. The calculated basal melting rate $M_b$ influences the velocity field according to Equation (2). The model in the present study forecasts temperature in the bedrock, and thus the GHF at the ice–bedrock interface has temporal variations. The bedrock is 3000 m thick divided vertically into 17 equal layers; constant physical parameters are used for the bedrock (density = 2700.0 kg m$^{-3}$, heat capacity = 1000.0 J kg$^{-1}$K$^{-1}$, and heat conductivity = 3.0 W m$^{-1}$K$^{-1}$), used in Parizek and Alley (2004).

We adopted different vertical resolution setups in computations of the temperature and age of the ice. The ice profile was discretized with 101 even vertical layers for thermodynamics; it was discretized with 2661 unevenly spaced vertical layers (finer near the base to resolve the thin layers of old ice) for age calculations, which was optimized following Saito et al. (2020). In the typical ice column thickness of 3000 m near DF, the vertical resolution was set to approximately 20 m near the surface and 20 cm near the bedrock, which is sufficient to resolve paleoclimate information

(glacial–interglacial annual layer variations) of ~1 ka. We used the rational function-based constrained interpolation profile (RCIP) scheme in the advection equation for the numerical scheme, as in Saito et al. (2020). One significant advantage of this scheme is the avoidance of numerical diffusion and ability to reasonably preserve the time derivative of age, which is critical to the resolution of old ice. We have tested the sensitivity to the vertical resolution of temperature calculation and found that using fine vertical resolution leads to the formation of a temperature inversion layer in the bottom of the ice, which can be a significant error in estimating basal temperature gradient and basal melting. Therefore, we set the number of vertical layers of the model for thermodynamics as 100 (each approximately 30 m thick) to prevent the representation of temperature inversion layers. The time steps of the calculation of temperature and age were set to 20 years.

## 3. Model calibration using DF age and temperature profiles
### 3.1. Experimental design

This section applies the 1-D model to DF under a realistic climate history for model calibration and parameter constraint. Parrenin et al. (2007) determined the $p$-value as ~3.7 for DF, but the chronology of ice older than 335 ka BP was not established at that time; therefore, we revisited DF to determine the $p$-value covering the entire DF ice core age–depth dataset. The glaciological boundary conditions at DF are summarized in Table 1: we used an ice thickness of 3028 m, a present-day SMB of 30 ice equivalent mm $a^{-1}$ (equivalent to 27.3 freshwater mm $a^{-1}$, based on Kameda et al., 2008 and Fujita et al., 2011), and a mean ice surface temperature at present of −55.5 °C. We determined the boundary condition of ice surface temperature by calibrating the temperature profile to be consistent with measured temperature profiles of the top 500 m of the ice, within uncertainty ranges of the observations. The observed present-day 10-m-depth annual mean snow temperature is −57.3 °C (Kameda et al., 1997), which was also used in Parrenin et al. (2007). We note that the annual mean surface air temperature (SAT) based on meteorological observations was −54.4 °C during the period 1995−1997 (Yamanouchi et al., 2003).

The model was forced by a realistic history of SAT and SMB. We used local SAT anomalies at DF for the past 715 ka BP (Uemura et al., 2018) and the benthic record of marine oxygen isotope data (Lisiecki and Raymo, 2005) to construct a continuous time series of SAT anomalies during the last 2 Ma. We applied a simple translation of $\delta^{18}$O to scale the temperature change at DF by the amplitude of glacial−interglacial cycles:

$$\Delta Ts = \alpha(\beta - \delta^{18}\text{O}), \quad (10)$$

where $\delta^{18}$O is the benthic marine oxygen isotope value [‰]; we set $\alpha = 4.5$, and $\beta = 3.23$ to scale the amplitude of the glacial cycles, which generated a time series of temperature change over the last 2 Ma, as shown in Fig. 3a. We used past SMB as a function of temperature anomaly compared with the present day following Huybrechts and Oerlemans (1990), which is based on saturation vapor pressure:

$$a(t) = a(ref) \cdot \exp\{22.47[\frac{T_0}{T_f(ref)} - \frac{T_0}{T_f(t)}]\}\{\frac{T_f(ref)}{T_f(t)}\}^2, \quad (11)$$

where $a(t)$ and $a(ref)$ represents past and present SMB rates, respectively. $T_0 = 273.16$ K is the triple point of water, and $T_f$ is the atmospheric temperature above the inversion layer as a function of surface temperature ($T_f$ [K] $= 0.67 T_s$ [K] $+ 88.9$). From this function, an increase in surface air temperature of 1 °C increases SMB by approximately 7% (Fig. 3b). At the Last Glacial Maximum (LGM, approximately 20 ka BP), when SAT was 8 °C cooler, the SMB was approximately 60% of that of the present day, which is consistent with reconstructions based on the isotopic content of the ice (Parrenin et al., 2016). This relationship between SAT and precipitation changes used herein was within uncertainties estimated from observations and climate model simulations, following a summary by IPCC AR6 (Chapter 9.4.2.3; Fox-Kemper et al., 2021), which used the studies of Bracegirdle et al. (2020) and Frieler et al. (2015). Although this relationship is not based on SMB,

but rather on precipitation, herein we assume the precipitation change ratio is the same as that of the SMB. The other boundary conditions (ice thickness and GHF) were set as constants in the present study.

We used a result of transient simulation obtained by a 3-D ice sheet model IcIES, which computes dynamics and thermodynamics of ice sheets using the shallow-ice approximation to simulate past ice thickness history. The experimental design was similar to that of Saito and Abe-Ouchi (2004, 2010) with some changes; the domain of the 3-D model was the whole Antarctic continent, and the horizontal resolution was set to 32 km. The spatial distribution of the GHF was from Martos et al. (2017). The model was initialized using the present-day condition, and forced by the same temperature and SMB changes as those of the 1-D model forcing for the past 2 Ma (Fig. 3a). The migrations of the grounding lines were not forecasted, instead the positions of grounding lines were fixed to the present day. We note that the advancement of grounding lines during glacial periods has a minor impact on the ice thickness, in particular around the DF region, compared with the changes in climate forcing (Saito et al., 2010). We extracted the history of changes in the ice thickness at DF and EDC, which showed that the ice thickness was reduced by ~200 m during glacial periods, mainly because of reduced SMB (Fig. 3c).

Using this set of boundary conditions, we conducted simulations with different GHFs (50–70 mW m$^{-2}$) to calibrate the model with observed values at the DF ice core. We used the depth–age profile of the DF ice core, which was constructed by orbital tuning of a gas record above ~2500 m, and by matching to the AICC2012 chronology below that depth (Kawamura et al., 2017). We also used the measured depth–temperature profiles from the JARE54 surveys conducted during the 2012–2013 Antarctic summer (Buizert et al., 2021). The model was initialized with the conditions of 2 Ma BP, where the initial age and temperature were set to 0 years and −10 °C, respectively, for the entire ice column. All experiments were integrated for 2 Ma to reach the present day; therefore, the age of any ice older than 2 Ma did not appear in the experiments. These simplified initial conditions generated unrealistic temperature fields in the early stage of the simulation, but realistic glacial cycle forcing prevailed over the entire ice column within approximately 100 ka. Therefore, we mainly analyzed the results of the last 1.5 Ma, which is sufficient to discuss old ice in this study. Furthermore, we also applied this model to the conditions at EDC to check whether the model could simulate the observed temperature and age profiles at this location (Table 1).

We also conducted three sensitivity experiments to investigate the impacts of the $p$ parameters, uncertainty in the amplitude of past temperature changes, and inclusion of past ice thickness changes, respectively. We found that $p = 3$ gave one good age profile when compared with the ice-core data; hence, we set $p = 3$ as the reference in Sect. 3. The uncertainty in the past temperature change was based on a study that proposed that the temperature change at the LGM in interior regions of the East Antarctic ice sheet was less than previously estimated (Buizert et al., 2021). We conducted a set of experiments where SAT anomalies were set to 0%, 25%, 50%, and 75% of the standard experiments, while keeping changes in SMB the same.

| Parameters | DF | EDC |
|---|---|---|
| Ice thickness [m] | 3028 | 3233 |
| Surface mass balance rate [ice equivalent mm a$^{-1}$] | 30.0 | 28.4 |
| Surface temperature [°C] | −55.5 | −54.65 |

**Table 1**: List of parameters used in Sect. 3. Ice thickness (DF and EDC), surface mass balance rate, and surface temperature at EDC come from Parrenin et al. (2007); surface mass balance rate at DF comes from Kameda et al. (2008) and Fujita et al. (2011); surface temperature at DF is calibrated in this study and is within previously observed ranges (Kameda et al., 1997; Yamanouchi et al., 2003).

**3.2. Results for DF**

In Fig. 4, the simulated temperature profiles at 0 ka (end of the simulations) with different GHFs under the same $p$-value ($p = 3$) are compared with observations (Fig. 4a). The close-up of the bottom 120 m of the ice column is shown in Fig. 4b; the basal temperature was well below melting point with a GHF of 54 and 56 mW m$^{-2}$, and at the melting point with a GHF > 58 mW m$^{-2}$. Compared with the observed temperature profile (Fig. 4, black lines), the simulated temperature near the ice base was colder by approximately 1 °C. In all simulations, the simulated temperature profiles were generally colder than observed temperature profiles, especially in the middle of the ice columns (Fig. 4a). The generally colder temperature of the ice may have several explanations. One is related to the pressure melting point of the ice. We used a pressure melting point of ice that depended only on local pressure, but there is also a dependence on the impurities and air content of the ice (e.g., Parrenin et al., 2017; Passalacqua et al., 2017). A second explanation is related to the uncertainty in vertical velocity of the ice parameterized with $p$ because a larger vertical advection contributes to a colder ice temperature.

The time series of simulated basal ice melting rates over the last 500 ka show that there have been significant temporal changes in these rates over time (Fig. 5a). With a GHF of 54 mW m$^{-2}$, the temperature at the ice base has been below the melting point through the last 500 ka. In contrast, in the case of a GHF of 56 mW m$^{-2}$, the basal melting rate is zero at 0 ka, while the maximum basal melting rate of 1 mm a$^{-1}$ occurs at the later stages of interglacial periods (e.g., 100 ka BP). This temporal variation in basal melting rate is caused by glacial-cycle forcing in SAT and SMB, and minimum basal melting tends to occur at the end of glacial periods as it lags SAT. This result is broadly consistent with previous studies (Saito and Abe-Ouchi, 2004; Van Liefferinge et al., 2018) in that colder ice, which accumulated during glacial maximums, advects towards the ice base owing to an increased SMB during interglacials. A larger GHF ($\geq$ 60 mW m$^{-2}$) results in basal melting occurring most of the time, with an increase in basal melting rate of approximately 1 mm a$^{-1}$ for every 5 mW m$^{-2}$ increase in GHF.

The simulated age profiles at the present day are compared with the ice core-based profiles in Fig. 6a. With a small GHF (54 mW m$^{-2}$) where basal melting does not occur, the ice age at the ice–bedrock interface is > 1.5 Ma. In contrast, if basal melting occurs, the ice age at the ice–bedrock interface can be much younger; for example, 761 or 620 ka for a GHF of 60 or 62 mW m$^{-2}$, respectively. The result obtained with a GHF of 60 mW m$^{-2}$ exhibits the closest fit to the data in terms of the age of ice at the base of the ice column. In this article, we define the "resolution of age" (ka m$^{-1}$) as the inverse of annual layer thickness as an indicator of sufficient resolution for the chemical and isotopic contents of the ice (Lilien et al., 2021). In Fig. 6b, the resolution of old ice is compared with the actual DF ice core. The model results largely reproduced the glacial–interglacial contrasts in annual layer thickness caused by the temporal variations of SMB at this locality. The observed resolution of age was approximately 0.5–1 ka m$^{-1}$ near the base of the ice core, and the model results using a GHF of 60 mW m$^{-2}$ reproduced similar values. Furthermore, in a scenario with no significant basal melting, the annual layer thickness of 1.5 Ma BP ice is approximately 0.1 mm because 1.5 Ma ice appears directly above the bedrock (Fig. 6b, dark blue lines). In accordance with the results described above, a larger GHF tends to result in a higher basal melting rate and younger age of ice at the base of the column. One critical point is that an excessive GHF (i.e., an increase of the order of 2 mW m$^{-2}$) can have a considerable effect on the age of the ice and the likelihood of old ice.

### 3.3. Results for EDC

We also applied this model using the conditions at EDC to enable performance checks at an additional location. We used the parameters listed in Table 1 and conducted sensitivity experiments with different GHFs. For the vertical velocity profile, we used $p = 2.3$ following Parrenin et al. (2007). The model generally resulted in colder temperatures compared with observations, similar to that found at DF (Fig. 7). We note that the pressure melting point of the ice depended only on local

pressure in Fig. 7, but several studies have considered the pressure melting point of the ice as a function of the pressure and air content of the ice, which has shown that the basal temperature is at the pressure melting point (Buizert et al., 2021). Modeling using a GHF of 56 mW m$^{-2}$ gave a basal ice age of approximately 800 ka (Fig. 8a), which is similar to the value (802 ka) presented in Veres et al. (2013), and the resolution of age closely fits the chronology estimated from ice-core analysis (Fig. 8b). One important result is that the threshold of GHF that allows basal melting is 4 mW m$^{-2}$ lower at EDC compared with DF. This lower GHF can be attributed to the combination of larger ice thickness, smaller SMB, and higher SAT at the present day. The estimated GHF at EDC is smaller than that given by Parrenin et al. (2017), who estimated it to be 60 mW m$^{-2}$. This difference can be attributed to the difference in the history of basal melting, or the application of past climate history derived from DF to EDC. The results from the application of our model to EDC suggest that it may be applicable to different glaciological conditions, particularly different ice thicknesses and SMBs.

### 3.4. Sensitivity to vertical velocity profiles, temperature amplitudes, and ice thickness changes

Next, we evaluated the sensitivity of the temperature and age profiles to different vertical velocity profiles, temperature amplitudes, and ice thickness changes over glacial cycles. In Fig. 9, results using different $p$-values under an identical GHF (60 mW m$^{-2}$) are compared. A larger $p$-value induced a lower basal melting rate because of a larger vertical velocity and downward advection of cold ice from the surface, although this only had a minor impact on the temperature profile. The simulated age profiles indicate that a larger $p$-value induces a younger age of ice at mid-depths within the ice column (Fig. 9b), which is also a result of a larger vertical velocity. The age of the ice at the base of the column was approximately 800 ka BP in all five of the variable $p$-value simulations, partly because of the compensating effects of greater advection and less basal melting.

The results using DF conditions with different amplitudes of temperature change but constant GHF and $p$ parameters (GHF = 60 mW m$^{-2}$ and p = 3) are summarized in Fig. 10. Here, we changed the $\alpha$-value in Equation 10 (1 is the control case). In the smallest amplitude experiment ($\alpha = 0$), the temperature was set to the interglacial level and did not change in time. Note that the SMB variation was the same in all sensitivity experiments. The control experiments exhibited colder ice temperatures near the middle of the ice column compared with observations, and this cold bias was reduced when a smaller temperature amplitude over the glacial cycles was used (Fig. 10a), broadly consistent with Buizert et al. (2021). A smaller amplitude of the glacial cycle resulted in a younger age of ice at the bottom of the ice column (Fig. 10b) because of larger basal melting rates (Fig. 10c). This is because the mean temperature over the glacial cycles increases if the temperature amplitude of glacial–interglacial cycles is reduced. The results using a fixed surface temperature ($dTs = 0.0$) corresponded to the same present-day SAT for the last 2 Ma, which induced basal melting of ~1.5 mm a$^{-1}$ during most of this time. A slight fluctuation in basal melting still occurred owing to time-dependent SMB.

The results without ice thickness changes did not impact temperature profiles at the present-day (Fig. 11a), but impacted the history of basal melting (Fig. 11c). The mean basal melting rates at constant GHF can be reduced if ice thickness changes are included because the reduced ice thickness during glacial periods decreases the pressure-melting point. Moreover, the inclusion of ice thickness changes affects the phase of basal melting rates because it reflects the reduction in ice thickness and pressure-melting point at the base of the ice during glacials. The minimum in basal melting during the last glacial cycle occurs at the end of the LGM in the control experiment; in contrast, it occurs at the present-day in the no ice thickness change scenario. The absence of ice thickness changes results in larger mean basal melting rates and a younger age of ice at the base of the ice column (Fig. 11b). These results suggest that the basal melting rate in the past can be larger than the present-day rate.

### 3.5. Summary of Sect. 3

On the basis of the results described in this section, we conclude that using a combination of

$p = 3$ and GHF = 60 mW m$^{-2}$ gives reasonable temperature and age profiles. Therefore, we decided
to use these values as calibrated parameters for the DF region; this was performed for the following
reasons. Later in the article, we investigate the possibility of old ice in the DF region using different
parameters of ice thickness and GHF because glaciological surveys have suggested that there are
spatial variations in these parameters (e.g., Carson et al., 2013). Hence, obtaining precise tuning at
one specific DF location is unnecessary. In this study, we calibrated the GHF under a vertical
velocity profile of $p = 3$, but calibrating the model with the combination of an uncertain GHF and
vertical velocity profile is possible. According to the age profile, the results with $p = 3$ may not
necessarily be the best because the simulated age profile tends to underestimate the age of ice,
particularly 500 m above the bedrock. Therefore, we do not state that a GHF of 60 mW m$^{-2}$ is a
single best estimate for the DF location compared with previous estimates (Burton-Johnson et al.,
2020; Talalay et al., 2021) because there were assumptions made in the vertical velocity profiles and
experimental design of this study. Furthermore, the calibrated GHF has some dependence on the
uncertainty in temperature and ice thickness changes over the glacial cycles.

## 4. Sensitivity studies using various parameters around DF
### 4.1. Experimental design
This section investigates the impact of the three parameters, ice thickness, SMB, and GHF,
which may have spatial variations in the DF region. We investigated a range of ice thicknesses
between 2000 and 3200 m, based on an ice thickness map of the area around DF (Fig. 1). We used a
present-day SMB range of 25–35 ice equivalent mm a$^{-1}$. There is large uncertainty in GHF; we
adopted a range of 50–70 mW m$^{-2}$. The list of experiments is given in Table 2. Other aspects of the
experimental design are the same as in Sect. 3.

| Variable | Parameter range |
|---|---|
| Ice thickness [m] | 2000–3200, every 100 |
| Present-day SMB rate [ice equivalent mm a$^{-1}$] | 25–35, every 1 |
| GHF [mW m$^{-2}$] | 50–70, every 2 |

Table 2: List of experiments in Sect. 4.

### 4.2. Results
In Fig. 12a, the relative effects of ice thickness and GHF on basal temperature are compared,
using a constant SMB (30 mm a$^{-1}$). As in Sect. 3, we used an ice thickness of 3028 m, which is
comparable to that at DF, and a GHF for basal melting of 60 mW m$^{-2}$. On the basis of the gradient of
contours in Fig. 12a, an increase in ice thickness of 100 m has a comparable impact on the basal
temperature as does an increase in GHF of 2 mW m$^{-2}$. In Fig. 12b, the relative effects of ice
thickness and SMB are compared using a constant GHF (60 mW m$^{-2}$). A larger SMB results in a
colder temperature; a 10% change in GHF leads to a ~4 °C change in the basal temperature, while a
10% change in SMB leads to a ~1 °C change. These results are generally consistent with those of
Fischer et al. (2013), and suggest that the spatial distribution of SMB (~20% for the DF area) has a
minor impact on the basal temperature compared with that of the ice thickness.
We further investigated the impact of different ice thicknesses on age profiles using the
climatic conditions at DF (SMB = 30 ice equivalent mm a$^{-1}$) and a calibrated GHF (60 mW m$^{-2}$).
Figure 13a shows the simulated age of the ice at 50 and 100 m above the ice–bedrock interface,
which were used as indicator depths for potential coring sites by Fischer et al. (2013). The results
indicate that the rate of aging of ice decreases with ice thickness between 2800 and 3200 m owing to
the occurrence of basal melting. Note that the age of 2 Ma BP is the limit of the experiments, and the
results indicate that the old ice exists 50 m above the bedrock if the ice thickness is thicker than
~2100 m. Figure 13b shows the age resolution of the 1.5 Ma BP ice, indicating that a larger ice
thickness tends to show a finer age resolution. The vertical age profiles and resolution of ice ages at

five selected ice thicknesses with constant GHF are shown in Fig. 14. According to Figure 14b, the expected age resolution of 1.5 Ma ice is approximately 10 ka m$^{-1}$ with an ice thickness of 2800 m, and 20 ka m$^{-1}$ with a smaller ice thickness of 2200 m.

## 5. Application to the DF–NDF transects

### 5.1. Experimental design

In this section, we apply the 1-D model to interpret the internal layers of the ice near DF under idealized boundary conditions. Here, we used the dataset from 17 December 2017 obtained by ground surveys during JARE59 (2017–2018), which comprises a 54 km-long transect from DF to NDF (Fig. 1). The horizontal axis of Fig. 15 indicates the distance from DF, and the vertical axis indicates the depth from the surface. The gray shading indicates the reflectivity, which is an indicator of contours representing ice of the same age. The bedrock elevation, shown by brown lines, was detected based on the maximum reflectivity from the base (Tsutaki et al., 2022). The bedrock elevation was calibrated to match the observed bedrock elevation at DF. We calculated the 1-D age and temperature profiles of the ice at approximately 400 m intervals along the transect. We assumed that the vertical profile of vertical velocity could be determined locally, and that there were no horizontal interactions in temperature and age in this simulation. The present-day SMB was linearly interpolated between DF (30 ice equivalent mm a$^{-1}$) and NDF (25.5 ice equivalent mm a$^{-1}$). Note that the estimated SMB at NDF is 13% smaller than that at DF based on shallow ice cores (Oyabu et al., 2023). Because only very limited information on the spatial distribution of GHF is available, we set a uniform value of 60 mW m$^{-2}$ following the discussion in Sect. 3. As described in Sect. 3, the initial age of the ice was set to 0, the temperature set to −10 °C, and the model was integrated over the last 2 Ma of forcing until it reached the present day (Fig. 3).

### 5.2. Results

In Fig. 15, the computed vertical profiles of the age are overlaid on a radargram using seven selected ages (colored lines), and the simulated basal temperature is indicated by shading in the bottom panel. The colored bar below the radargram indicates the simulated present-day basal temperature. The simulated distribution of ice age captured large-scale features in the black–white contour lines derived from the radargram signal (grayscale color in Fig. 15). The simulated age contours of 21 ka BP (approximately 500 m depth) and 128 ka BP (approximately 1500 m depth) can be traced from DF, although the deepest horizon corresponding to an age older than 300 ka BP is hard to see in this image. Where ice is relatively thick (e.g., 20–25 km from DF), the simulated age of the ice at the ice–bedrock interface is younger than 700 ka BP, while ice older than 1.5 Ma BP occurs where the ice is relatively thin. On the basis of the results shown in Fig. 13b, we note that thin ice gives a poorer age resolution for the old ice. A comparison of the simulated ice age and radargram signal gives an opportunity to examine the validity of the model results. For example, between 5 and 35 km from DF, the computed 128 ka BP contour deviates to shallower levels (by 150 m) compared with the traced horizon for the age obtained from the radar measurements, suggesting that the model overestimates the age of the ice near the bedrock in such locations.

## 6. Discussion

In this study, we used a 1-D ice-flow model, which computes the temporal evolution of age and temperature profiles. We used glaciological conditions at DF to tune some unknown parameters according to the observed temperature and age profiles. The results showed that the age profile is sensitive to the choice of GHF, but one experiment using a specific combination of GHF and vertical velocity profile exhibited reasonable temperature and age profiles (Figs 4 and 6). One important result is that the melting rate at the base of the ice exhibits temporal changes associated with glacial–interglacial forcing. This is caused by relatively cold ice deposited during glacial periods being pushed towards the bottom of the ice column by increased SMB and downward advection

during interglacial periods, as shown in previous studies (e.g., Van Liefferinge et al., 2018). This point is critical for preserving old ice because basal melting rates during past interglacials can be higher than that of the present day (Fig. 5). Our sensitivity experiments highlighted the relative effects of ice thickness and GHF, whereby a small GHF excess above the condition that induces basal melting can result in a considerable reduction in the age of ice at the ice–bedrock interface (Fig. 6a). Below, we discuss the limitations of the interpretations of our results, their relevance to previous ice-flow modeling studies, and uncertainty factors.

On the basis of data presented in Fig. 6, a GHF of 60 mW m$^{-2}$ sufficiently explains the observed temperature and age–depth profiles of the DF ice core. However, there is considerable uncertainty in the estimation of the actual GHF value at DF because of some simplifications in the model experiments and limited representations in physics. One point of difference is that the model tends to give a generally colder temperature profile compared with the observations (Fig. 4), which suggests that the model overestimates the GHF threshold of basal freezing. One possible reason for this difference is that the basal melting of ice can occur within a certain ice thickness; the extrapolation of observed temperature profiles at DF and EDC (Figs 4 and 7, black lines) shows that the ice reaches the pressure-melting point approximately 30 m above the bedrock. This feature cannot be simulated in the model of the present study, which assumes that basal melting can only occur at the ice–bedrock interface. These representations in the physics of basal melting can be improved by using enthalpy as a state variable and adopting polythermal ice sheet models (e.g., Aschwanden et al., 2012). There is also uncertainty in the parameterization of the conductivity and heat capacity of the ice. We use these parameters as a function of temperature, but they can depend on the fabric of the ice, which makes it challenging to estimate them. Hence, these physical parameters can be a source of uncertainty in estimating GHF, and can be a source of difference from other studies. Another important factor in the temperature profiles is the temperature anomaly over glacial cycles, as a smaller glacial–interglacial temperature change tends to result in a warmer, more linear temperature profile compared with the control experiment (Fig. 10a). The surface air temperature change over the last glacial cycle used in this study is based on deuterium and oxygen isotopes (Uemura et al., 2018), which exhibit an LGM temperature anomaly of approximately 8 °C (Fig. 3a). A recent study proposed that the temperature anomaly at the LGM at DF and EDC was approximately half of previous estimates based on the observed temperature profiles and other independent methods (Buizert et al., 2021). This study is in agreement with Buizert et al. (2021) in that our control experiment exhibits colder ice temperatures, especially at mid-depth within the ice column, and a smaller temperature difference between glacial and interglacial periods improves the modeled temperature profiles (Fig. 10a). If this is indeed the case, the actual threshold of the GHF value for basal freezing should be lower than that used in the control experiment. We also found that if the temperature anomaly is half that of the control case, a GHF smaller than the control value (58 mW m$^{-2}$) gives the closest age profile. We investigated the sensitivity to ice thickness as in Fig. 13, and obtained comparable results in terms of the age near the bottom of the ice column (not shown). These results indicate that several uncertainties (e.g., climate forcing and vertical velocity) can affect the temperature and age profiles under a certain condition, but if we calibrate the GHF with the DF ice-core age profile as in Sect. 3, we obtain comparable results regarding the sensitivity to ice thickness.

We note that the simulated age of the ice depends on the shape of the vertical velocity profile of the ice. The formulation of the present study uses a smaller vertical ice velocity, especially near the base, compared with that used in Fischer et al. (2013). Because the age of the ice is related to the inverse of the vertical velocity, a different vertical velocity profile or $p$ parameter can lead to a quantitatively different result. Moreover, vertical velocity profiles represented by a single $p$-value are merely one assumption; this formulation is derived from a solution of an idealized ice-sheet configuration (Lliboutry, 1979), which may not be the case for realistic ice sheets. For example, the observed magnitude of layer thinning of the DF ice core exhibits a decreasing trend over the bottom

500 m (Fig. 6). According to analyses of the DF ice core (Azuma et al., 1999; Saruya et al., 2022) or 3-D ice sheet modeling (Seddik et al., 2011), deformation of the ice or flow regime towards the bottom of the ice is complex, suggesting parameterizing vertical velocities is difficult particularly near ice bottom. Improving velocity fields in ice sheet model would be an important issue for future studies

We also note that the resolution of 1.5 Ma ice depends on ice thickness. In particular, Lilien et al. (2021) presented similar 1-D ice-flow model results from BELDC (Beyond EPICA Little Dome C, ice thickness of ~2765 m) constrained by radar-imaged internal layers and estimated the resolution of 1.5 Ma ice as $19 \pm 2$ ka m$^{-1}$. Our results for EDC conditions (with a small enough GHF to keep the base of the ice frozen) have an ice age resolution of approximately 10 ka m$^{-1}$ (Fig. 8, dark blue lines), which is approximately half that of Lilien et al. (2021). This difference can be attributed to the combination of model parameters, such as ice thickness, $p$ of the vertical velocity profile, or SMB history (3233 m and $p = 2.3$ in this study), because the two studies adopted the same formulation of the vertical velocity profile. According to Figs. 13 and 14, a larger ice thickness leads to a better resolution of the ice age if the base of the ice remains frozen throughout time. It is worth mentioning that the approach in ice thickness are different between Lilien et al. (2021) which used ice thickness of 2765 m, including a thickness of a basal unit of ~200 m and thus an effective ice thickness of 2565 m. Therefore, the different effective ice thickness (3233 m for EDC) would be the most critical factor for the difference of the age resolution of 1.5 Ma ice when compared with Lilien et al. (2021), who used BELDC conditions.

Application of the 1-D model to the transect between DF and NDF provides an opportunity to examine the influence of spatially varying glaciological conditions (e.g., ice thickness and GHF) on the age of the ice. The simulated age–depth distributions with constant GHF but different ice thickness and SMB exhibit general agreement with observed internal horizons (Fig. 15). One noticeable model–data discrepancy occurs at 14–18 km from DF, where the simulated age contours of 128 ka BP are ~150 m above the isochrone horizons traced from DF. This model–data discrepancy indicates that the effects of vertical or horizontal advection (Huybrechts et al., 2007; Sutter et al., 2021), or spatial variation of GHF may have contributed to this difference. Although the relative importance of the spatial distributions of GHF, SMB, and horizontal flow is difficult to assess in the present study, we expect that future glaciological data constraints and model developments will better constrain these uncertain parameters and the spatial distribution of old ice. One recently published present-day SMB from the vicinity of the DF region exhibits spatial variabilities reflecting surface topographical features (Van Liefferinge et al., 2021). On the basis of systematic sensitivity experiments (Sect. 4), we have shown that the impact of SMB on the age of the ice is relatively minor compared with that of ice thickness, but the small-scale features present in internal reflection horizons of the ice can be improved by using the spatial distribution of present-day SMB, and this will contribute to the selection of the most suitable drilling site.

## 7. Conclusions

We draw the following conclusions from this study:

1. In experiments using the DF configuration, the model largely reproduced the observed age and temperature profiles under a calibrated GHF. If the GHF is small enough to keep the basal temperature below the melting point, it is expected that ~1.5 Ma ice could be present. According to Figs. 14 and 15, the simulated annual layer thickness of ~1.5 Ma ice is approximately 0.05 to 0.1 mm, which corresponds to 10 to 20 ka m$^{-1}$. According to IPICS, this is a feasible resolution for analysis with minimized effects of diffusion. This is also true for EDC, but the threshold of GHF for basal melting is different because of a different ice thickness and SMB.

2. Under the configuration and range of parameters of the present study, the ice thickness has a larger impact on basal melting than does the present-day SMB; an ice thickness difference of ~100 m corresponds to an SMB difference of 5 ice equivalent mm a$^{-1}$ (Fig. 12). Near the DF

region, the ice thickness exceeds such a spatial variability, while SMB does not. Although there is considerable uncertainty in the spatial distribution of GHF, ice thickness is suggested to be one of the most critical factors for the preservation of old ice.

3. The calibrated GHF in this study, which is based on an ice-core age profile, has uncertainties. The basal melting rate, which is critical to the age of ice near the bottom of the column, is determined by the thermal conditions. The basal melting exhibits temporal variability as a result of glacial–interglacial changes in climate, and the maximum basal melting tends to occur at the end of interglacials. Thus, the basal melting is influenced by climate forcing of past temperature and ice thickness changes, which have uncertainties. Furthermore, a vertical velocity profile parameterized with a uniform $p$-value can be a source of uncertainty, and may have a limited ability to represent complex ice flow near the bottom of the ice column.

4. From the simulation of the DF–NDF transect, a small ice thickness and colder basal temperature are the necessary conditions for the presence of old (~1.5 Ma) ice. However, a small ice thickness contributes to a coarser resolution of the old ice (small annual layer thickness), which may make it difficult to extract paleoclimate information on glacial-interglacial times scales. As discussed in Pattyn (2010), ice thickness is found to be a compromising factor in the selection of a drilling site.

5. The simulation along the DF–NDF transect does not reproduce the depth of the internal layers of the ice corresponding to 128 ka BP at some locations (e.g., at distances 5–35 km from DF), suggesting a possible error in the simulated age of ice near the bottom of the ice column. The simulated age of ice in this area, especially where there is a large discrepancy between the simulation and radar images, could be caused by uncertainties derived from several assumptions or uncertainty in the model or methods, including spatial distributions of GHF, representation in vertical temperature profile that depends only on normalized height (DF ice core suggests complex ice-flow near its base), representation in thermodynamics associated with basal melting, or history of surface temperature changes. Therefore, future improvements in numerical models and methods would contribute to better constraining the age of the ice.

A recent compilation of ice thickness data around DF indicates the presence of complex and steep terrain in the area, with uncertainty in bedrock elevation of > 60 m (Tsutaki et al., 2022), highlighting the necessity of a high-spatial-resolution survey of bedrock topography. The results from this study help to support the interpretation of observational data and the selection of a suitable drilling site.

**Code availability:**
The numerical model is available from Github. https://github.com/saitofuyuki/icies2.git

**Data availability:**
The scripts and data for conducting experiments and analyzing results are available at AORI-CESD (https://cesd.aori.u-tokyo.ac.jp/cesddb/publication/index.html). All figures were generated using GMT version 4.5.9. The ice core chronology and temperature at DF are available from previously published articles (Veres et al., 2013; Kawamura et al., 2017; Buizert et al., 2021).

**Author contribution**
T. O., A. A-O., and F. S. conceived the study, developed the numerical model, designed and carried out the experiments, and analyzed the results. T. S., S. F., K. K., and H. M. provided glaciological data from JARE surveys and contributed to the experimental design. T. O. prepared the manuscript with contributions from all co-authors.

**Competing interests**

The authors declare that they have no conflict of interest.

**Acknowledgments**
We would like to thank Frédéric Parrenin and two anonymous referees for their valuable comments, which have substantially improved our manuscript. We thank Kenichi Matsuoka, Brice Van Liefferinge, and Ralf Greve for their fruitful discussions. This research was supported by JSPS Kakenhi JP17H06104, JP17H06323, and JP18H05294. T. O., A. A-O., and F. S. were supported by JPJSBP120213203. F. S. was also supported by JSPS Kakenhi JP17K05664. The 3-d ice sheet model simulations were performed on the Earth Simulator 4 at Japan Agency for Marine-Earth Science and Technology (JAMSTEC). We thank David Wacey, PhD, from Edanz (https://jp.edanz.com/ac) for editing a draft of this manuscript.

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

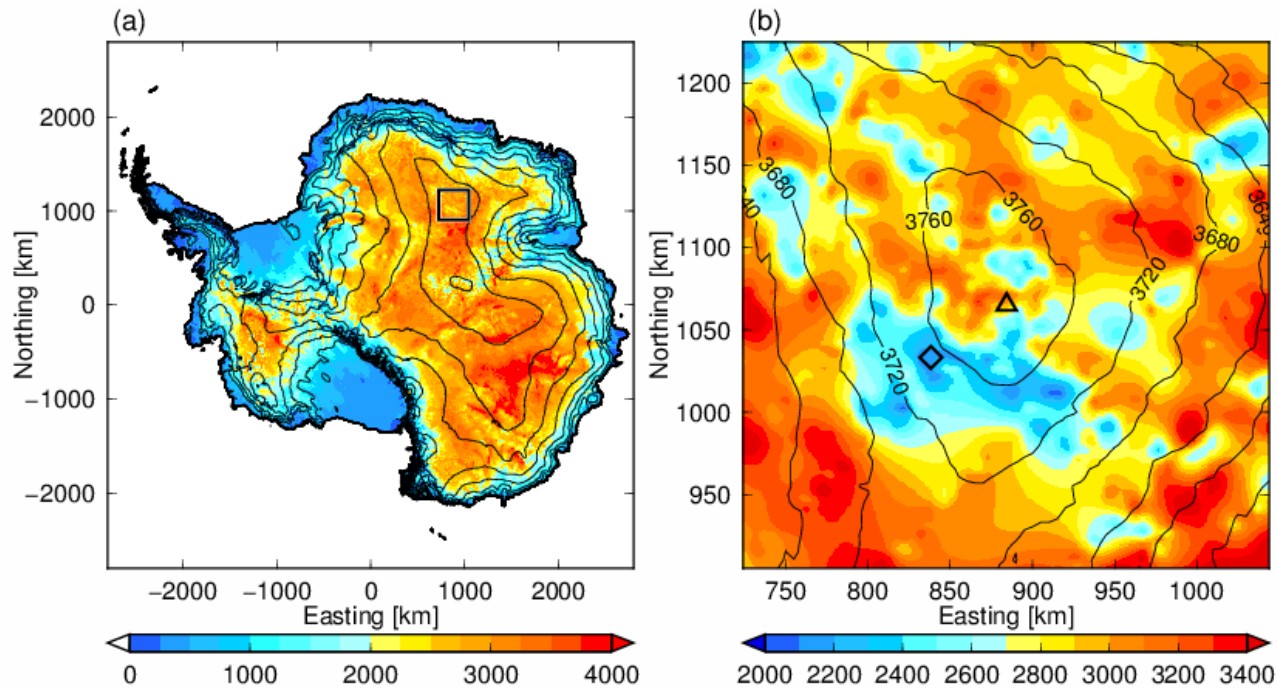

Fig. 1: (a) Map of Antarctica. The contours (every 500 m) indicate surface elevation, and colors indicate ice thickness, using BEDMAP2 (Fretwell et al., 2013). The square indicates the location of the inset shown in (b). (b) Enlarged view near DF (Dome Fuji). The triangle indicates the location of the DF ice core site, and the diamond indicates the NDF site.

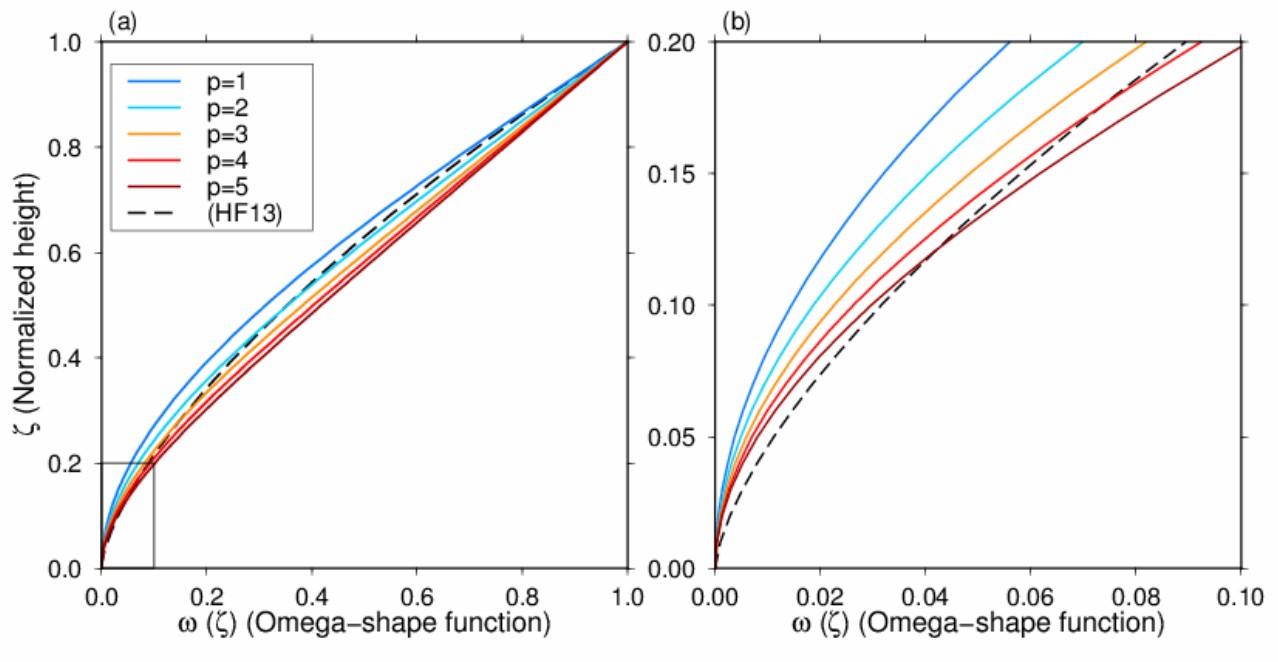

Fig. 2: (a) Normalized vertical velocity profiles adopted from Equation [3] with different $p$ parameters. The dashed black line (HF13) indicates the vertical velocity profile used in Fischer et al. (2013) with $m = 0.5$. (b) Enlarged view near the bottom of the ice column (see black rectangle in (a)).

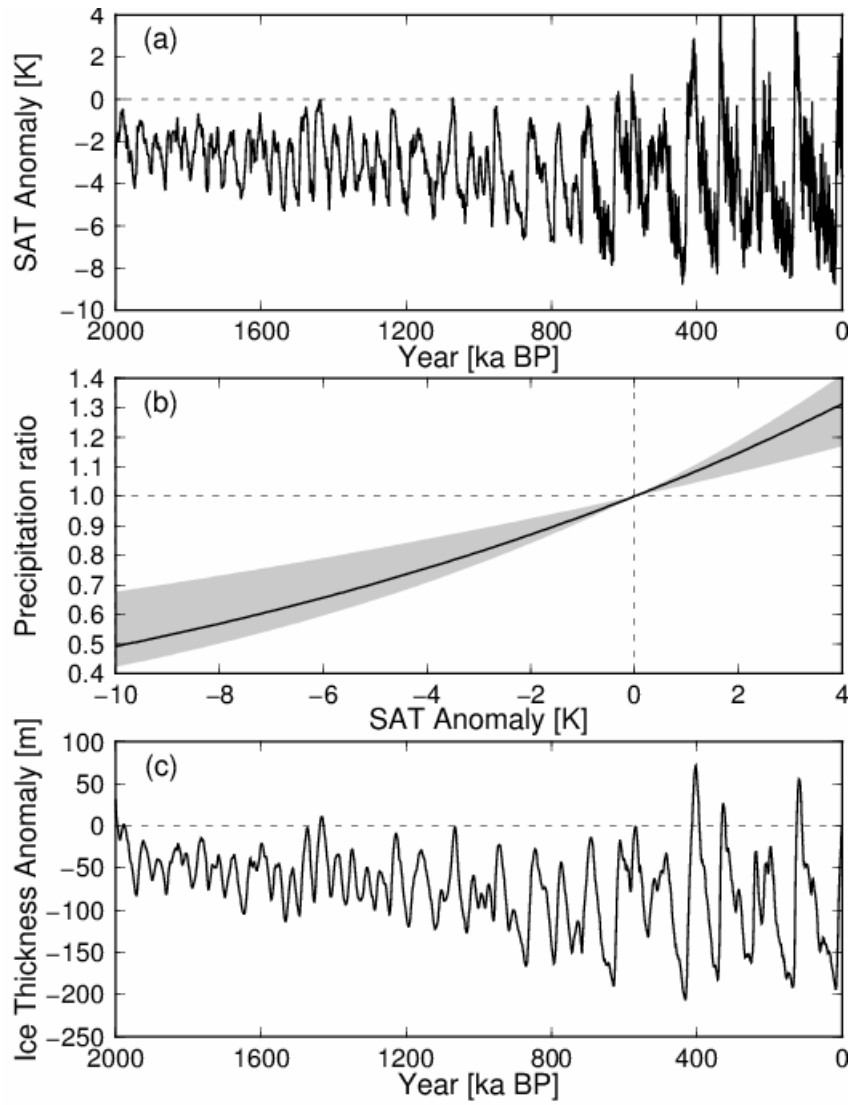

Fig. 3: Glacial cycle forcing used in the present study. (a) Surface air temperature (SAT) anomaly from the present day for the last 2 Ma. (b) Relationship between SAT anomaly and precipitation ratio. The black line corresponds to the relationship used in the present study; the gray shading indicates a 4%–9% increase per degree, summarized in Fox-Kemper et al. (2021). (c) Ice thickness anomaly at DF from a 3-D ice sheet model in the present study.

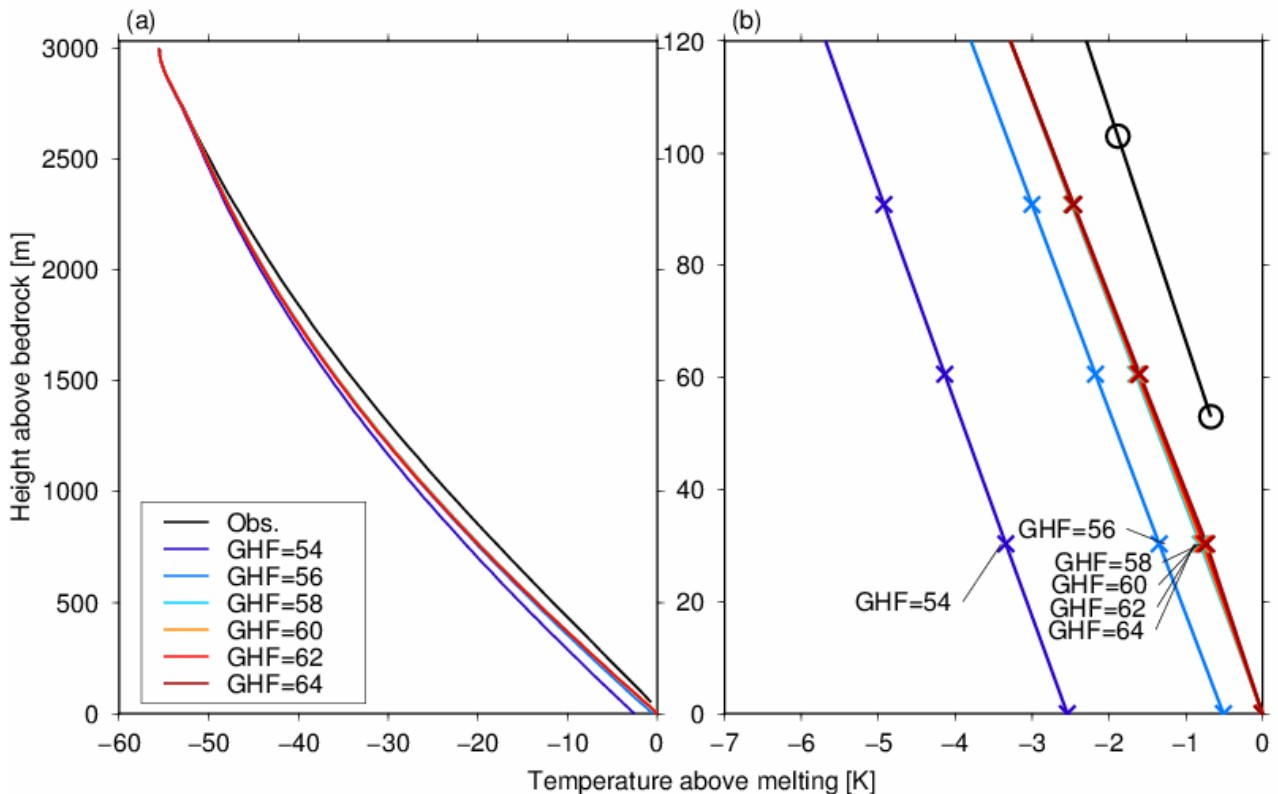

Fig. 4: Simulated vertical temperature profiles under the DF configuration (Table 1) with different geothermal heat fluxes (GHF; units are mW m$^{-2}$). (a) Simulated temperature profiles at 0 ka (end of the simulation) from the surface to the base. (b) Close-up of (a) for the bottom 120 m of the ice column. The black lines represent the measured temperature profiles and the black circles in (b) indicate the location of data points, while the colored crosses in (b) represent the model grid points.

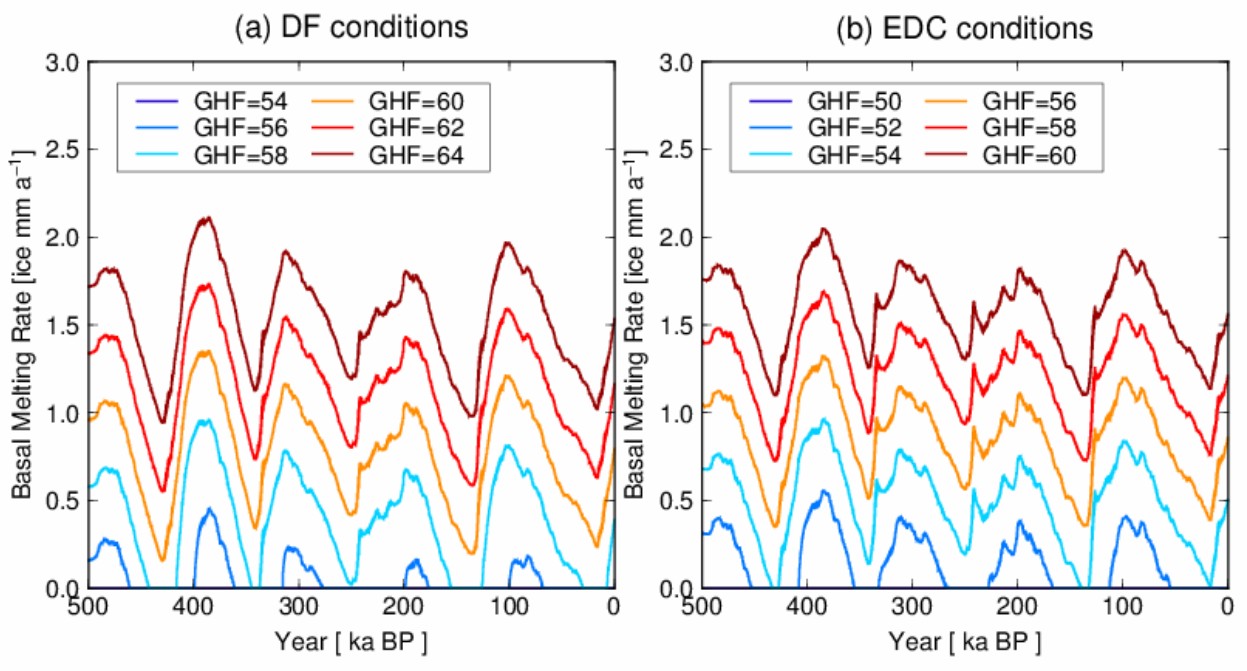

Fig. 5: Time series of the simulated basal melting rates of the last 500 ka under the DF and EDC configurations (Table 1) with different geothermal heat fluxes (GHF; units are mW m$^{-2}$).

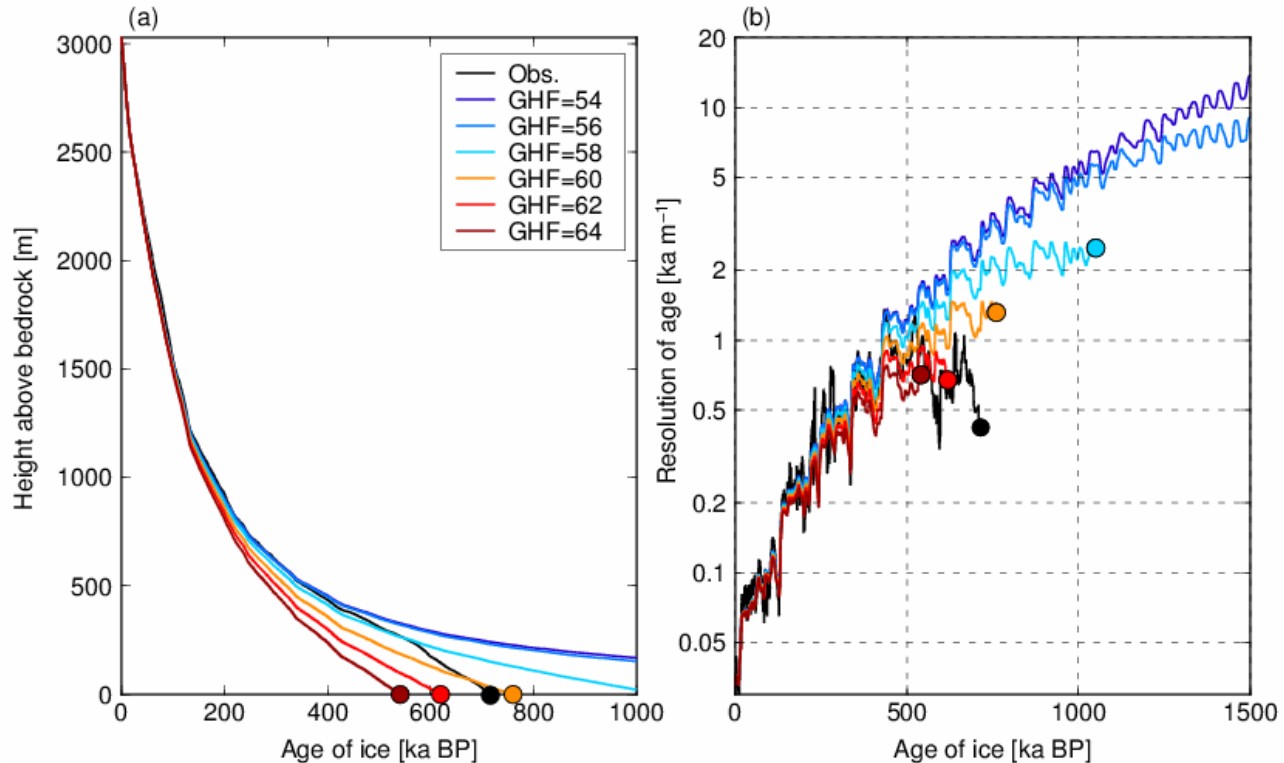

Fig. 6: Simulated vertical ice age profiles under the DF configuration (Table 1) with different geothermal heat fluxes (GHF; units are mW m$^{-2}$). (a) Vertical age profiles at present (0 ka). The black line represents the reconstructed depth–age profile based on the AICC2012 chronology (Kawamura et al., 2017). The circles indicate the bottom of the ice. (b) Vertical resolution of ice age, calculated by the central difference using the simulated vertical age profiles of (a).

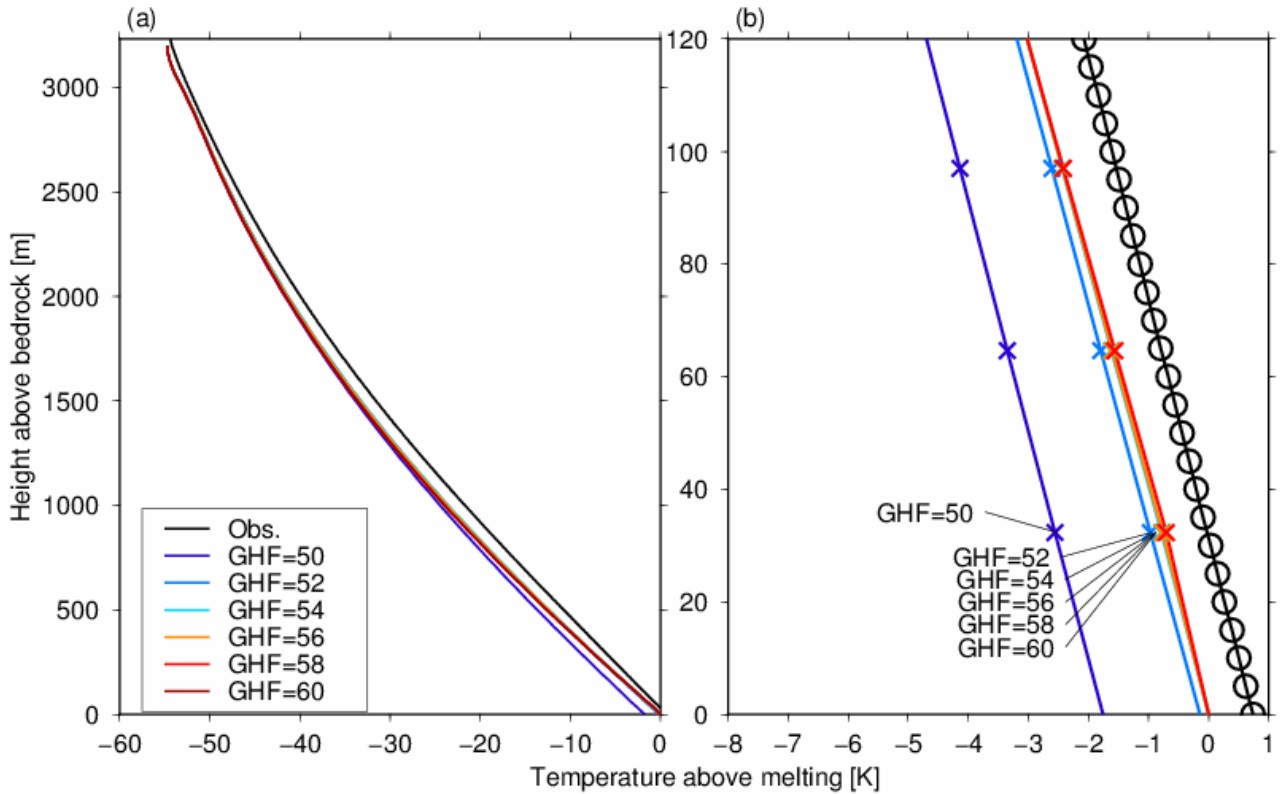

Fig. 7: Same as Fig. 4, but under the EDC configuration (Table 1) with different geothermal heat
fluxes (GHF; units are mW m$^{-2}$). The black lines represent the measured temperature profiles and the
black circles in (b) indicate the location of data points, while the colored crosses in (b) represent the
model grid points.

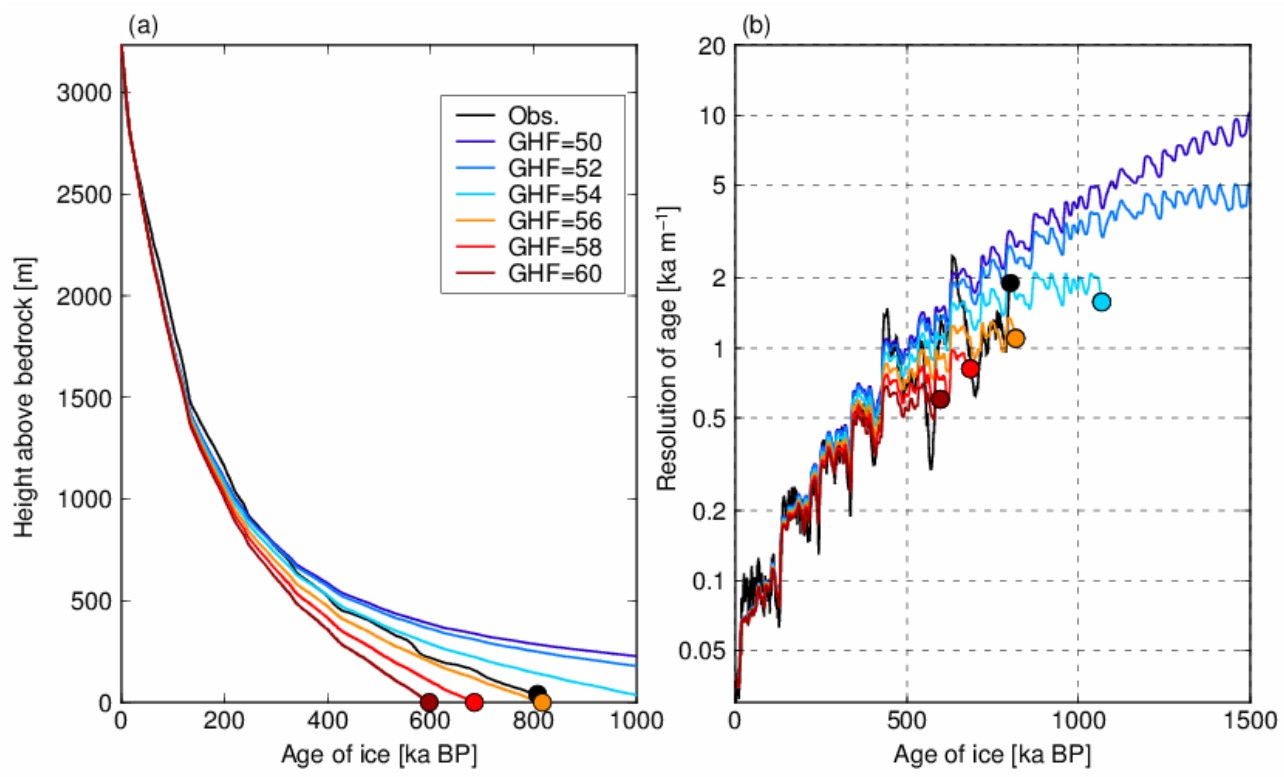

Fig. 8: Same as Fig. 6, but results under the EDC configuration (Table 1). The AICC2012
chronology (Veres et al., 2013) is used in this figure for the observed depth–age profile.

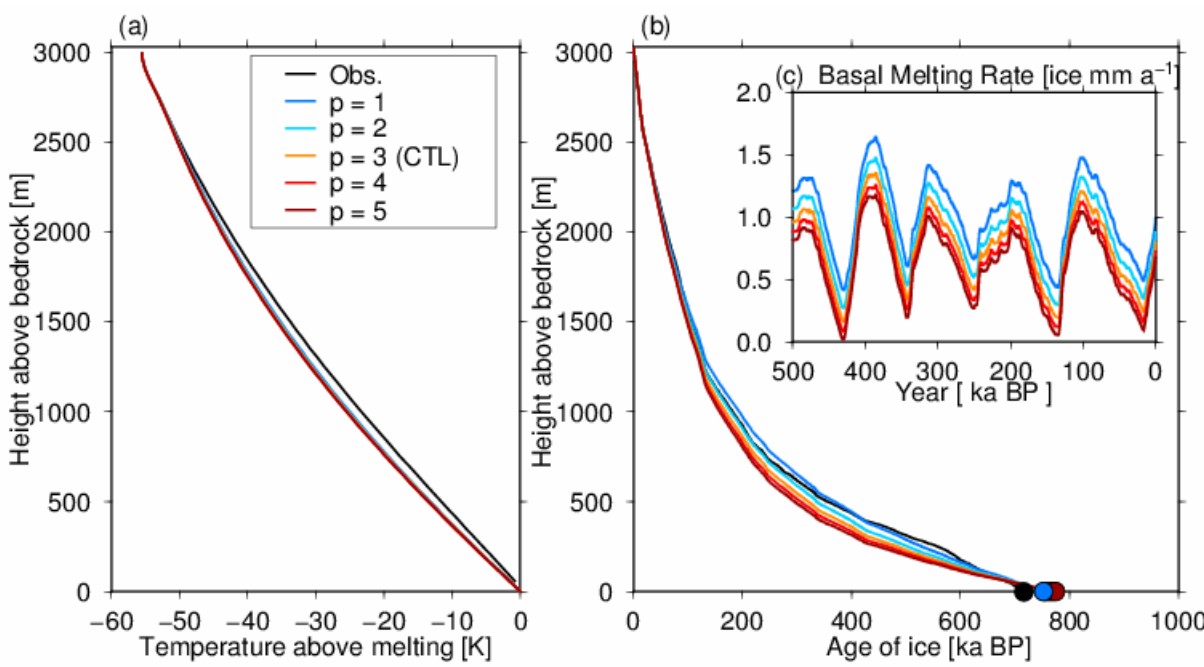

Fig. 9: Results of the DF configuration (Table 1) with different $p$ parameters. (a) Simulated
temperature profiles at present (0 ka) from the surface to the base. (b) Vertical age profiles at present
(0 ka). (c) Time series of basal melting rates over the last 500 ka. A geothermal heat flux of 60 mW
m$^{-2}$ is adopted in these experiments.

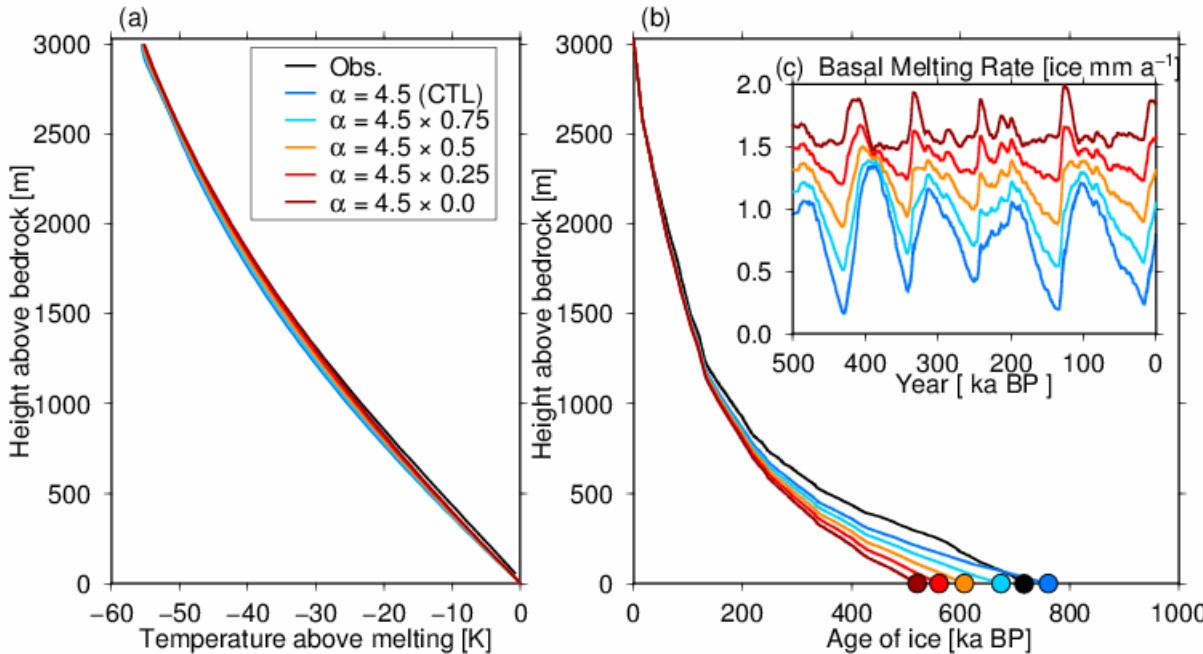

Fig. 10: Results of the DF configuration (Table 1) with different temperature amplitudes over glacial
cycles in Equation 10. A combination of $p = 3$ and GHF $= 60$ mW m$^{-2}$ is adopted in these
experiments. (a) Simulated temperature profiles at present (0 ka) from the surface to the base. (b)
Vertical age profiles at present (0 ka). (c) Basal melting rates of the last 500 ka.

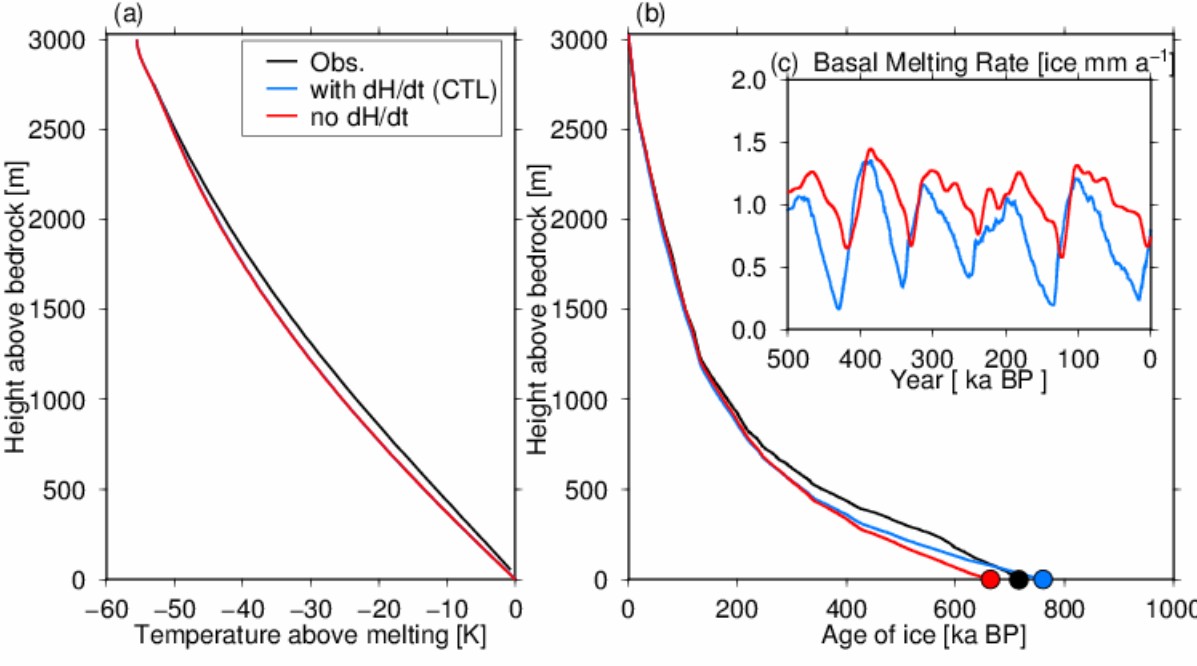

Fig. 11: Results of the DF configuration (Table 1) with and without ice thickness changes in the past.
A combination of $p = 3$ and GHF $= 60$ mW m$^{-2}$ is adopted in these experiments. (a) Simulated
temperature profiles at present (0 ka) from the surface to the base. (b) Vertical age profiles at present
(0 ka). (c) Basal melting rates of the last 500 ka.

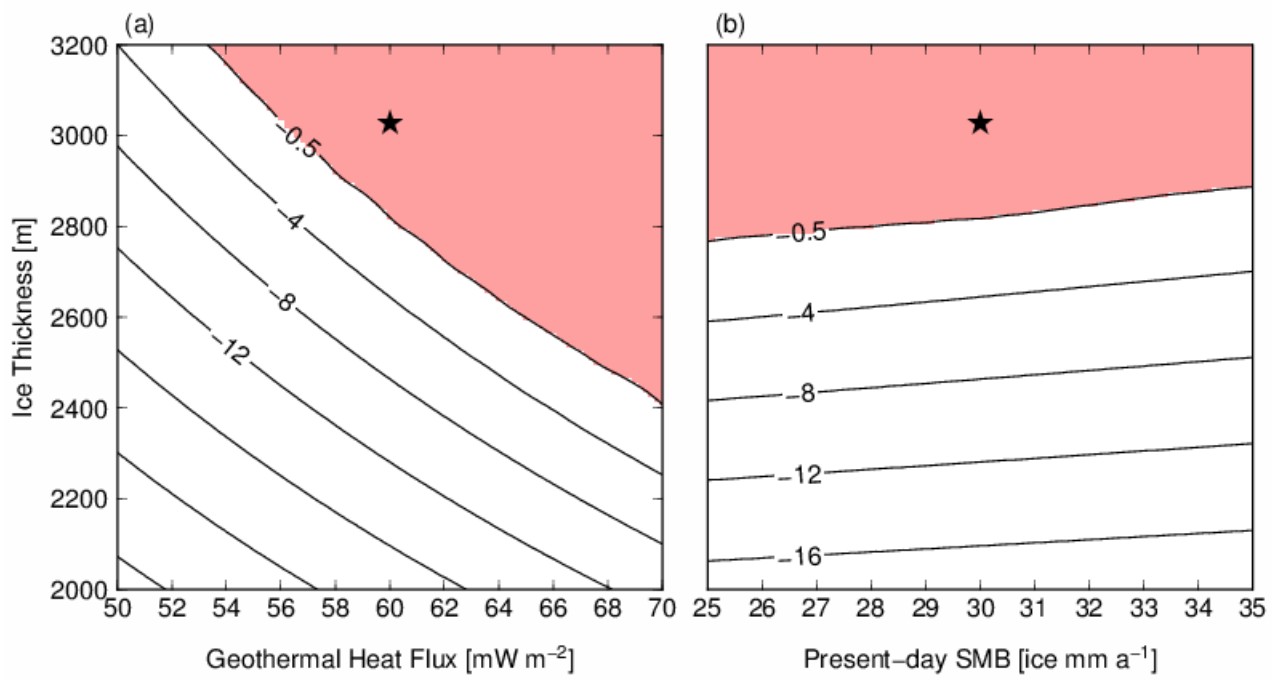

Fig. 12: Simulated basal temperature at the present day with combinations of ice thickness, geothermal heat flux, and present-day SMB. (a) Red shading indicates a basal temperature −0.5 °C below the pressure-melting point. (b) Basal temperature at the present day with GHF = 60 mW m$^{-2}$. The black star represents the condition at the DF ice core (H = 3028 m, SMB = 30 ice mm a$^{-1}$), with a calibrated geothermal heat flux (60 mW m$^{-2}$).

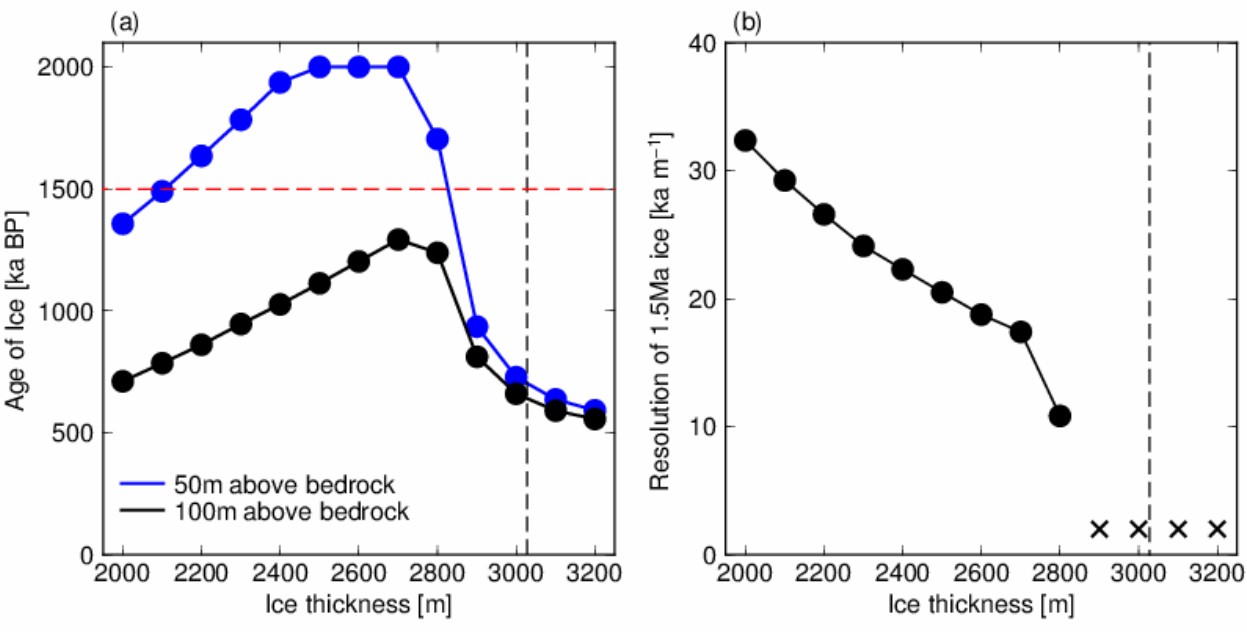

Fig. 13: Results with different ice thicknesses at the DF configuration (SMB = 30 ice equivalent mm a$^{-1}$ and GHF= 60 mW m$^{-2}$). (a) The black and blue lines indicate the simulated ages of the ice at 100 and 50 m above the bedrock, respectively. The vertical dashed line (H = 3028 m) indicates the condition at DF, and the horizontal red dashed line indicates the age of 1.5 Ma. Note that an age of 2 Ma is the limit of the experiments. (b) The vertical axis indicates the resolution of the ice age (ka m$^{-1}$) at 1.5 Ma BP. The crosses indicate that the 1.5 Ma age of ice does not exist under these

conditions.

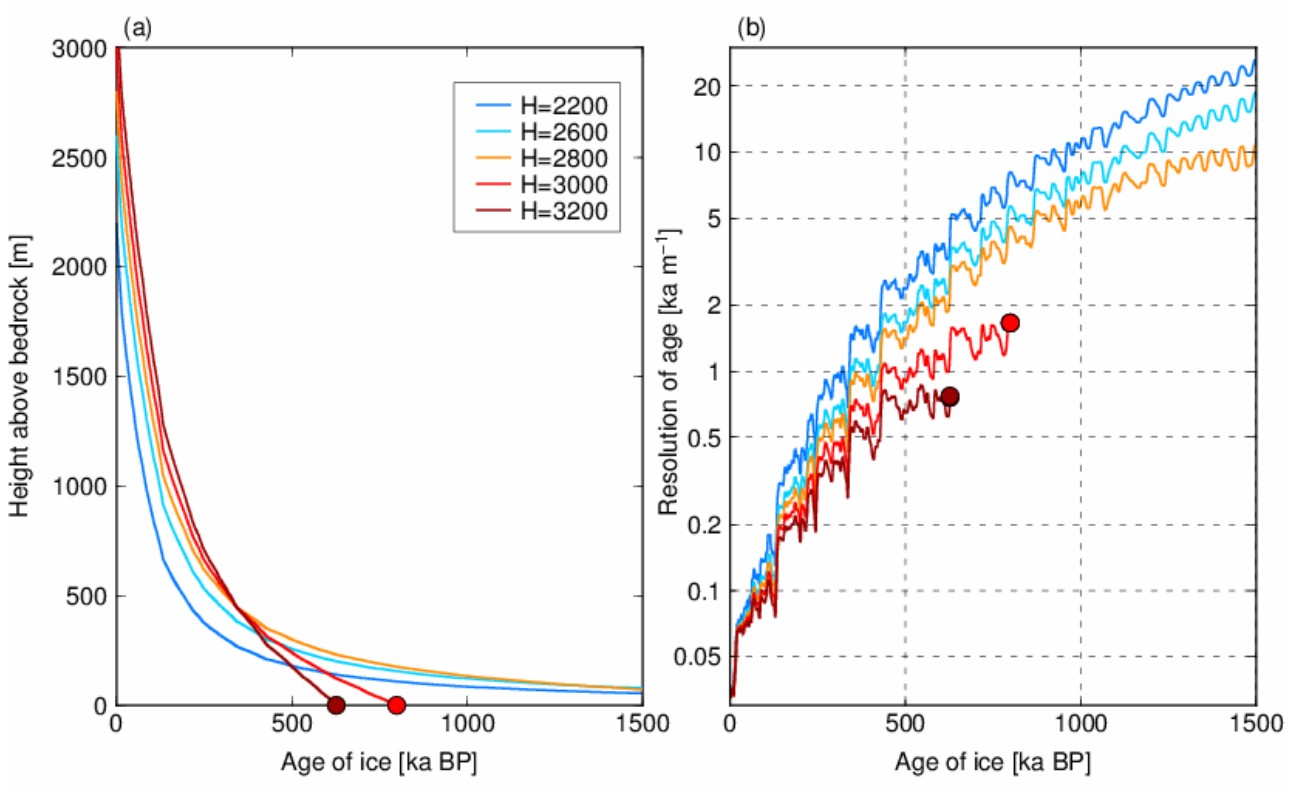

Fig. 14: Results with different ice thicknesses (2200, 2600, 2800, 3000, and 3200 m) and calibrated geothermal heat flux (60 mW m$^{-2}$) and SMB (30 ice equivalent mm a$^{-1}$) at DF. (a) Vertical age profiles at present (0 ka). (b) Vertical resolution of the ice age.

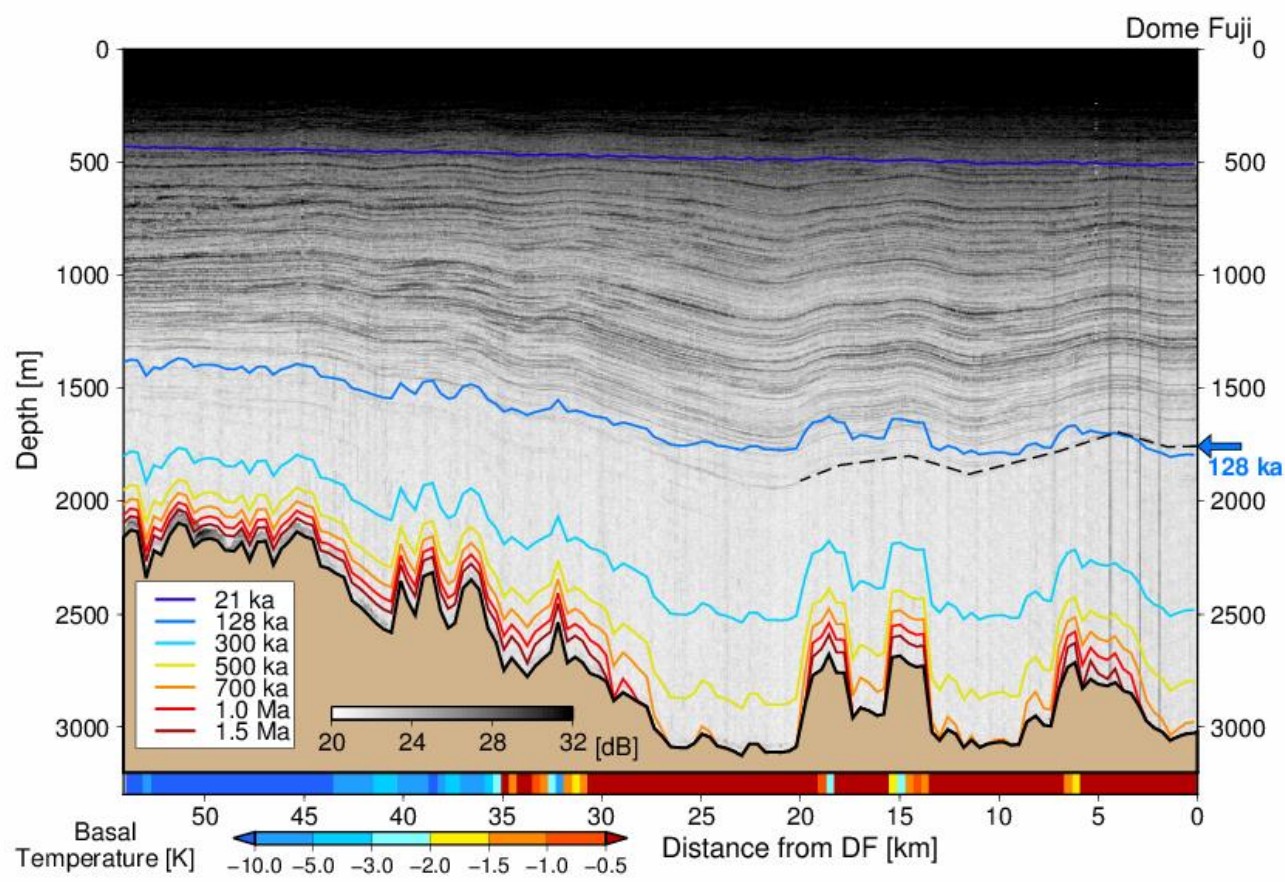

Fig. 15: Results of the experiments overlaid with the observed radargram for the DF–NDF transect. A combination of $p = 3$ and GHF = 60 mW m$^{-2}$ is adopted in these experiments. The horizontal axis indicates the distance from DF (km), and the vertical axis indicates the depth from the surface (m). The gray coloring indicates the reflection intensity from the ground radar surveys, and the color contours indicate the simulated age of the ice using the 1-D model. The black dashed line indicate the traced isochrone horizon from DF, corresponding to ~ 128 ka. The bottom color bar indicates the simulated basal temperature (relative to the melting point) at the present-day.