# Peer review of "A one-dimensional temperature and age modeling study for selecting the drill site of the oldest ice core near Dome Fuji, Antarctica"

_The Cryosphere, 2022_

## Author Comment (AC1)

We thank all three reviewers who provided precise and valuable feedback on our manuscript. Our complete response is in this document. The reviewer's comments are quoted in italic, and our answers follow. The revised sentences of the manuscript are indicated in red text. We will be happy to submit a revised manuscript that reflects these changes.

In the revision of the manuscript, we have applied two changes in the model experiments regarding the comments by reviewers.

First, we are replacing the numerical method of the ice-flow model in estimating basal temperature gradient in calculating basal melting rates, as suggested by reviewers #2 and #3. The revised manuscript will use a one-sided difference discretization method at the ice-bed interfaces in estimating basal temperature gradient instead of the central difference method used in the original manuscript.

Second, we include the ice thickness differences associated with the glacial-interglacial cycle, as suggested by reviewer #1. These two changes led to a lower basal melting rate in the same conditions compared to the methods used in the original submission (Revised Figures 4 and 5). Because of the smaller basal melting rate, the calibrated geothermal heat flux at DF and EDC conditions can be larger by ~5 mW m$^{-2}$ than in the original submission (Revised Figure 6). This modification in the model impacts quantitative results regarding the possibility of old ice at given conditions, as summarized in a sensitivity experiment with different ice thicknesses (Revised Figure 13).

For now, we have not revisited all experiments, but we have revisited representative experiments used in the article. We show the revised result figures in the top of at the top of this response letter (Revised Figures 4-6, 10, 13). When we submit the revised manuscript, Figures 4-15 and manuscripts will be revised accordingly.

One note on ice thickness change is that in this response letter, we have used a simple model used in Saito et al. (2020), assuming that the steady-state ice thickness expected from the surface mass balance and the response time of ice thickness. This simple model with the paleoclimate forcing of DF shows ~200 m variations in ice thickness for the DF case (Figure S1 black lines). However, this amplitude is more significant than the results of a 3-dimensional ice sheet model (Saito and Abe-Ouchi 2010), which exhibits ~150 meters at DF (Figure S1 red lines). We think the ice thickness change of 200 meters might be a bit large. In this response letter, we show the results using ice thickness changes from the simple model. In the revised manuscript, we will revisit all experiments using ice thickness changes from the 3-d models extending the simulation to the past 2 Ma (Saito and Abe-Ouchi (2010) conducted past 220 ka simulations).

We have one minor change in the method of showing the resolution of age (annual layer thickness) in Figs. 6, 8, 10, 13, 14. The submitted article used the central difference of the simulated vertical age profiles to calculate age resolution. As one strength of the RCIP scheme is solving the spatial derivatives of the age simultaneously, the revised manuscript will use the simulated vertical derivatives of the age as the resolution of age. Although the difference from changing the method has a slight change in the result figures, we think showing the output from the RCIP scheme will be better.

[Figure]

Revised Figure 4: Simulated vertical temperature profiles under the DF configuration (Table 1) with different geothermal heat fluxes (GHF; units are mW m$^{-2}$). (a) Simulated temperature profiles at 0 ka (end of the simulation) from the surface to the base. (b) Close-up of (a) for the bottom 120 m of the ice column. The black lines represent the measured temperature profiles and the black circles in (b) indicate the location of data points, while the colored crosses in (b) represent the model grid points.

[Figure]

Revised Figure 5: Time series of the simulated basal melting rates of the last 500 ka under the DF and DC configurations (Table 1) with different geothermal heat fluxes (GHF; units are mW m$^{-2}$).

[Figure]

Revised Figure 6: Simulated vertical ice age profiles under the DF configuration (Table 1) with different geothermal heat fluxes (GHF; units are mW m$^{-2}$). (a) Vertical age profiles at present (0 ka). The black line represents the reconstructed depth–age profile based on the AICC2012 chronology (Kawamura et al., 2017). The circles indicate the

bottom of the ice. (b) Vertical resolution of ice age.

[Figure]

Revised Figure 10: Simulated vertical ice age profiles under the EDC configuration (Table 1) with different geothermal heat fluxes (GHF; units are mW m$^{-2}$)

[Figure]

Revised Figure 13: Results with different ice thicknesses, under GTH=60 mW/m$^{-2}$. (a) The black and blue lines indicate the simulated age of the ice at 100 and 50 m above the bedrock, respectively. The vertical dashed line (H = 3028 m) indicates the condition at DF, and the horizontal red dashed line indicates the age of 1.5 Ma. Note that an age of 2 Ma is the limit of the experiments. (b) The vertical axis indicates the resolution of the ice age (ka m$^{-1}$) at 1.5 Ma BP. The crosses indicate that the 1.5 Ma age of ice does not exist

under these conditions.

Response to Frédéric Parrenin (REVIEWER #1)

*This manuscript presents simulations of a 1D age and temperature model, mainly for the Dome Fuji ice core and region, but also for the EDC ice core. The main aim of the manuscript, as the title reads, is to infer potential old ice drilling sites in the Dome Fuji region. The manuscript is generally of excellent quality. It is precise and reads well. However, I have a few suggestions for the authors which could further improve the relevance of the manuscript. I let the authors decide if they want to include these suggestions in their simulations, or simply discuss them in the discussion and outlook sections.*

Thank you for careful reading and giving us fruitful comments. We decide to adopt suggestions by conducting additional experiments and adding figures.

*Main comments:*
*- The model is interesting since it is a transient model, while other models used for the same purpose were steady (or pseudo-steady). However, the authors do not use the full power of this transient aspect of the model, since they fixed the ice thickness. As the authors wrote, the ice thickness is a primary parameter controlling the basal melting/temperature and therefore basal ice age. Therefore, a glacial-interglacial ice thickness change of 200 m can have an important impact on the simulations.*

Thanks a lot for suggestion. We use a simple model used in Saito et al. (2020), assuming that the steady-state ice thickness at the summit is proportional to the $1/(2n+2)$ power of the surface mass balance (n=3, n is Glen's flow law exponent), and the response time of ice thickness change (3000 years). This simple model with the paleoclimate forcing of DF shows ~200 m variations in ice thickness for the DF case (Figure S1, black lines). We conducted experiments with the inclusion of the evolving ice thickness. We found that the inclusion of the evolving ice thickness term tends to have smaller basal temperature than the fixed ice thickness case, probably because of reduced pressure-melting point during glacial periods leads to a colder temperature than fixed ice thickness experiments. One note is the ice thickness difference of ~200 meters is larger than that simulated from 3-dimensional ice sheet model (Saito and Abe-Ouchi 2010), which exhibits ~150 meters at DF and different response time (Figure S1, red line).

The revised manuscript will use evolving ice thickness, as well as a different method in calculating temperature gradient at ice sheet-bedrock interface (as indicated by

the top of the response letter).

[Figure]

Figure S1: Ice thickness anomaly for the last 250,000 years. Black lines represent a simple model used in Saito et al. (2020) with two different response thickness timescales. Red line represents 3-d ice sheet model results with transient climate forcing (Saito and Abe-Ouchi 2004; 2010). Note that the period of the 3-d model simulation is the last 220,000 years.

[Figure]

Figure S2: Basal melting rates for DF experiments, without (left panel) and with (right panel) evolving ice thickness.

*- The authors find a shift between observed and simulated temperature profile near the bed.*
*They reckon that this is due to polythermal ice, but there is another explanation. Indeed, the pressure melting point is not so well known. Apart from pressure, it also depend on*

*the impurities and air content of the ice. Catherine Ritz discussed that in a thesis 30 years ago, and this discussion is still relevant I think.*

We agree with this. We use pressure-melting temperature depends only on local pressure. Meanwhile, several studies use pressure-melting points with the function of pressure and air content. We will add sentences in the first paragraph of subsection 3.2. And also, we will add sentences in subsection 3.3 (EDC) because pressure melting point dependency on air content of the ice is frequently used in EDC.

(L247-248, section 3.2): In all simulations, the simulated temperature profiles were generally colder than observed temperature profiles especially in the middle of the ice columns (Fig. 4a). The generally colder temperature of the ice shift may be derived from several reasons. One is the pressure melting point of the ice. We use pressure melting point of the ice depending only on local pressure, but it also depends on the impurities and air content of the ice (e.g., Parrenin et al. 2017; Passalacqua et al. 2017). Another one is the vertical velocity of the ice parameterized with p, because a larger vertical advection contributes to a colder temperature of the ice.

(L290-291, section 3.3) The model generally results in colder temperatures compared with observations, similar to DF (Fig. 9). We note that we use the pressure melting point of the ice depending only on local pressure in Fig. 9, but several studies use the pressure melting point of the ice as a function of pressure and air content of the ice, which shows that the basal temperature is at the pressure melting point (Buizert et al. 2021).

*- The 1D simulations for EDC are not discussed as much as the simulations for DF.*
*I understand the authors have deeper interests for Dome Fuji, but I think it could make the manuscript more valuable if the EDC case is discussed more. For example, I would have been interested by a graph showing the basal melting variations at EDC with time.*

Thank you for suggestion. We will add time-series of basal melting in EDC case (Revised Figure 5).

*- On the contrary, I did not find the simulation along the DF-NDF transect so interesting. To make it really interesting, it would have been necessary to invert the parameters (in particular accu and GHF) to fit the observed isochrones. There is no reason to assume accu varies linearly and GHF is constant.*

While one motivation for applying the 1-D model to the DF-NDF transect is examining the drill site, the role of the DF-NDF transect experiment in the present study is an exercise experiment under an idealized setting. We will revise section 5-1

(experimental design) to clarify them.

*- The thermal parameters of the ice (conductivity, heat capacity) are not so well known. There are several parametrizations. Conductivity also depend on the fabric, which makes it even more challenging to estimate them. I think a discussion on these different parametrizations would have been valuable.*

We will write out the parameterizations of conductivity and heat capacity of the ice used in this study in the method section for clarification. We will insert discussion on the uncertainty in the thermal conductivity and heat capacity of the ice in section 6, the manuscript change appears below (L424-425).

$$\text{Heat conductivity:} \quad \kappa = 9.828e^{-0.0057T} \text{ W m}^{-1}\text{K}^{-1}$$
$$\text{Heat capacity:} \quad c_p = (146.3 + 7.253T) \text{ J kg}^{-1}\text{K}^{-1}$$

*- Catherine Ritz showed a long time ago that it is best to simulate the temperature variations in the bedrock. Indeed, temperature waves propagates in the upper continental crust and the geothermal flux at the ice-bedrock interface cannot be assumed constant with time.*

The model in the present study forecasts temperature in the bedrock, so the geothermal flux at the ice-bedrock interface has temporal variations. The bedrock is 3000-m thick with 17 even vertical layers. The constant physical parameters of the bedrock are used in this model (density=2700.0 kg m$^{-3}$, heat capacity=1000.0 J kg$^{-1}$K$^{-1}$, heat conductivity=3.0 W m$^{-1}$K$^{-1}$). We will clarify this in the revised manuscript in the model description section.

*- It would have been interesting to make a Monte-Carlo simulation for DF and EDC to see which sets of parameters are acceptable. Here, the parameters are changed one after the other but there are probably covariances.*

We agree that Monte-Carlo simulation has an advantage in estimating a good set of parameters for specific sites at DF and EDC. Nevertheless, we would like to keep systematic sensitivity experiments in the present article rather than precise tuning with the previous DF ice core. This is because this study focuses on the range of glaciological parameters around DF (ice thickness 2000-3200m, different SMB and GTH), which can differ from conditions from the previous DF ice core drilling site.

*Minor comments:*
*- l. 77-80: What is important for applying a 1D ice flow model is not the value of the*

*horizontal velocity, but how the ice flow parameters (e.g., ice thickness) varies upstream. For example, an ice flow line can be 100 km long with a surface velocity of 1 m/yr, and a 1D model could still be appropriate if everything is constant upstream.*

We agree with this. We will add that the small spatial variations in horizontal ice velocity are also a factor in applying 1D models.

"One-dimensional vertical ice-flow models have been used as the vertical profiles of age and temperature near Antarctic Domes, where horizontal flow is relatively minor. Horizontal velocity in the vicinity of DF and NDF is < 2 m a$^{-1}$, and it has minor spatial variations, evidenced by satellite-based measurements (Rignot et al., 2011, 2017; Mouginot et al., 2012)."

*- l. 97-98: Parrenin et al. (2017) did not exactly assume that basal melting was constant. They used the pseudo-steady assumtion, which states that temporal variations in basal melting are the same than temporal variations of surface accumulation rate.*

Thank you for correcting this. We will revise the sentences:

"Parrenin et al. (2007) assumed that basal melting rates were constant over time, and Fischer et al. (2013) used a constant climate forcing. Parrenin et al. (2017) used the temporal evolution of basal melting with a pseudo-steady assumption, assuming that the temporal variations in basal melting rates are the same as accumulation rates."

*- eq. (2) and l. 130-131: I think there is an inconsistency here. As eq. (2) is written, a positive value of Mb means ice refreezing, not melting.*

Yes, negative indicates ice melt. We will revise this.

*- l. 133-134 and eq. (3): This equation was first formulated in Parrenin and Hindmarsh (2007).*

Thanks a lot. We will refer to the article when describing the derivation of the formulation.

*- l. 145-146: Regarding the neglecting of heat production, I think it could justified by the small ice deformation near a dome (very low horizontal shear which is the dominant factor elsewhere).*

We agree with this. We will add that the relatively minor horizontal shear and small deformation near Antarctic domes is one reason for neglecting strain heating.

"The strain heating term is neglected in the present study, given that ice deformation would be minor near Antarctic domes because of very low horizontal shear."

*- l. 146-148: There are different parametrizations of ice conductivity and thermal capacity (see comment above). These are not discussed here, but I reckon they can have an important effect.*

We will address the uncertainty in parameterizing in the second paragraph of the discussion when discussing uncertainties in estimating geothermal heat flux. Please see our reply below (comments on L424-425, same topic)

*- l. 148: Is it not 917 kg/m^3 the standard value for ice density? (note the wrong unit in the manuscript).*

We use 910 kg m$^{-3}$ as the standard value for the density of ice sheet. This value is one frequently used value in 3-dimensional ice sheet models (e.g., Huybrechts and Payne 1996, EISMINT ice sheet model inter-comparison project).

*- eq. (5) and (6) assumes a constant geothermal heat flux, which is not the case since heat waves propagate in the upper continental crust (see comment above).*

As indicated above, the model in the present study forecasts temperature in the bedrock (3000-m thick bedrock with 17 even vertical layers), so the geothermal flux at the ice-bedrock interface has temporal variations. The prescribed geothermal heat flux is given at the bottom of the bedrock. We will revise the manuscript accordingly.

*- l. 164-166: I don't understand this sentence here. The formulation of the model does not allow for polythermal ice, so there is no reason to decrease the vertical resolution.*

We have tested the sensitivity to the vertical resolution of temperature calculation and found that the fine vertical resolution leads to the formation of temperature inversion layer in the bottom of ice. This can lead to a significant source of error in estimating basal temperature gradient and basal melting. We will change these sentences as this:

"Basal melting can occur in the interior of the ice as represented by polythermal ice sheet models. In contrast, the model in the present study assumes basal melting can occur only at ice-bed interface. We have tested the sensitivity to the vertical resolution of temperature calculation and found that the fine vertical resolution leads to the formation of a temperature inversion layer in the bottom of the ice, which can be a significant error in estimating basal temperature gradient and basal melting. Therefore in this article, for simplicity, we set the vertical layers of the model for thermodynamics as 100 (~30 meters) to prevent representing temperature inversion layers."

*- l. 203-205: Could you please write the equation relating SAT and accu? Is it the saturation vapor pressure relationship?*

Yes, the equation relating SAT and precipitation is from the saturation vapor pressure using temperature above the atmospheric inversion layer. We will add the formulation from Huybrechts and Oerlemans (1990) in the methods.

*- l. 216: I find it a shame that the ice thickness is fixed despite the model being transient (see comment above).*

Thanks a lot. In the revised manuscript, we use the ice thickness tendency term as the standard experiment, as well as changes in the calculation of basal temperature gradient. (response above)

*- l. 228-230: It would have been possible to initialize the age and temperature profile with steady profile, instead of constant values, for a faster convergence.*

We agree on this point. Meanwhile, assuming constant age and temperature is valid as the realistic glacial cycle forcing prevails over the entire ice column within approximately 100 ka. (This is indicated by L231-233)

*- l. 247-248: The obs-model temperature shift near the bed is probably due to the formulation of the pressure melting point (see comment above).*

We have addressed this above.

- l. 261-263: For sure! Without basal melting, the age is infinite at the base.

Exactly. As indicated later in the manuscript (L354-355 and Figure 13), the age of ice cannot exceed 2 Ma BP in this forward simulations initialized with the age of 0.

*- l. 296: Parrenin et al. (2017) also estimated the GHF at EDC (Figure 5c), but the value (~60 mW/m^2) is far higher than what you obtained here.*

Yes, in the original submission the calibrated geothermal heat flux from at EDC (51 mW m$^{-2}$) was significantly smaller than Parrenin et al. (2017). After receiving all reviewer's comments, we found that changing the method of calculating basal temperature gradient at ice-bed interface and the inclusion of the evolving ice thickness contributes to increase in geothermal heat flux necessary to reproduce EDC age profiles. We have revisited the EDC experiments, and we found that the geothermal heat flux at EDC would be 56 mW m$^{-2}$ according to Revised Figure 10, which is now closer to Parrenin et al. (2017)'s values. As indicated by the top of this response letter, change in

the method of estimating basal temperature gradient and consideration of ice thickness changes contribute to smaller basal melting. And the smaller basal melting led to an increase in geothermal heat flux necessary to account for observed age profiles at both of DF and EDC. There's still difference of ~4 mW m$^{-2}$ from Parrenin et al. (2017), this would be attributed to the difference in the history of basal melting, applying past climate of DF to EDC, or others.

*- l. 299: I find this paragraph a bit short (see comment above).*

We clarify the logic of the sentences. And we address the uncertainty of the pressure melting point of the ice, too.

"The results from the application to EDC suggests that our model may apply to different glaciological conditions, with different ice thickness and SMB."

*- l. 317-326: It would have been interesting to make a Monte-Carlo simulation to see which sets of parameters are acceptable (see comment above).*

We have addressed this above.

*- l. 364-379: This is a very simplified transect simulation (see comment above).*

We have addressed this above.

*- l. 424-425: Instead of using polythermal ice, use a different parametrization of pressure melting point.*

We address the uncertainty in parameterizing in the second paragraph of section 6 when discussing uncertainties in estimating geothermal heat flux. We will add sentences here.

"And also, there's uncertainty in the parameterization of the conductivity and heat capacity of the ice. We use these parameters as a function of temperature, but they can depend on the fabric of the ice, which makes it challenging to estimate them. Hence, we need to remember that these physical parameters can be a source of uncertainty in estimating geothermal heat flux and can be a source of difference from other studies."

*- l. 460-462: Of course, Lilien et al. find different results since they simulated BELDC and not EDC, with a very different ice thickness.*

Exactly, the ice thicknesses at EDC and BELDC are quite different. We will clarify that the ice thickness is one critical factor in the resolution of old ice from Lilien et al. (2021)'s results.

"Therefore, the different ice thickness (3,233 m for EDC) would be the most critical factor in the differences in the age resolution of 1.5 Ma BP ice with Lilien et al. (2021), who used BELDC conditions (ice thickness of 2,750 m). "
* * *
REVIEWER #2

*"A one-dimensional temperature and age modeling study for selecting the drill site of the oldest ice core around Dome Fuji, Antarctica" by Obase et al. details experiments utilising one dimensional age and temperature modeling of the Antarctic ice sheet. The validity of the model is demonstrated by comparisons with ice-core based age reconstructions and temperature measurements at both Dome Fuji and the EPICA Dome C core sites. Parameter sensitivity and selection studies for the Dome Fuji region are then conducted, and finally the optimised model applied to a ground-based radar survey in the region, and the simulated age horizons compared to isochrones from the radar survey. Overall, the paper is well written and easy to follow and is worthy of publication in the Cryosphere, after minor re visions as detailed below.*

Thank you for your careful reading and giving us fruitful comments. We address point-by-point replies to all of your comments below.

*Minor issues*
=============

*L83-85 The logic here isn't quite correct. While a lower accumulation rate is necessary to increase the number of years in a given thickness of ice, a lower accumulation rate will also reduce the vertical advection of cold from the surface down into the interior of the ice sheet, therefore increasing the temperature of the ice. So accumulation rate plays a dual and potentially competing role, but in terms of basal melt rates, lower accumulation is not necessarily a good thing.*

Thanks a lot for pointing out this. Exactly, a smaller accumulation rate contributes to a chance of basal melting by decreased vertical advection of ice, which was discussed by Fischer et al. (2013). We will revise the sentence:

"The key finding was that melting at the base reduces the likelihood of old ice, and the lower ice thickness than previous ice core sites are required conditions to avoid basal melting. And the smaller accumulation generally contributes to increasing the age of the ice at certain height from the bedrock, but also increase the chance of basal melting by the reduced vertical advection of the cold ice."

*L104-106 Parrenin et al 2017 (doi:10.5194/tc-11-2427-2017) applied a time varying rate factor to both the accumulation and melt rates in there 1-D modelling around EDC. This rate factor was based on variations from the EDC ice core for the last 800ka and was constant before 800ka.*

We revise the manuscript above (L97-99) by referring Parrenin et al. (2017) to state they use time-dependent basal melting.

"Parrenin et al. (2007) assumed that basal melting rates were constant over time, and Fischer et al. (2013) used a constant climate forcing. Parrenin et al. (2017) used the temporal evolution of basal melting with a pseudo-steady assumption, assuming that the temporal variations in basal melting rates are the same as accumulation rates."

And also we will revise the L104-106 as follows.

"Despite the close link between the temperature and age of ice owing to basal melting, the coupled simulations of thermodynamics and age of ice were not represented under transient climate forcing in previous modeling studies."

*L139-140 Need to make it clear that "m" is Fischer et al's equivalent to "p". Suggest re-wording from "in the case of m=0.5 in their study" to "where their parameter m fulfils a similar role to p in this study, the case of m=0.5"*
*L141-142 m=0.5 is only smaller than p=3 for zeta<0.3. Suggest re-wording from "with a smaller vertical velocity, particularly near the base of the ice" -> "with a smaller vertical velocity in the lower approximately third of the ice" or "with a smaller vertical velocity near the base of the ice"*

Thank you for your suggestions. We will revise the sentences:

"Compared with Fischer et al. (2013) with a different formulation of vertical velocity profile with m parameter (similar role with p of this study), m=0.5 (Fig. 2 dashed lines), p = 3 from Equation (3) gives a different vertical temperature profile, with a smaller vertical velocity near the ice of the base."

*L161 Are you really calculating the temperature gradient at ice-bedrock interface using a central difference? If so you would need to be modelling the temperature down into the bedrock. If you are doing this, you should mention that the thermal domain extends down into the bedrock and give the boundary conditions at the bottom of the rock domain. If you are only modelling the thermal domain in the ice, then you must be using a one-sided difference discretization at the ice-bedrock interface.*

Yes. The original manuscript use a central difference by extrapolating ice

temperature below bedrock. Meanwhile, as indicated by the top of this response letter, we decide to use one-sided difference discretization in estimating basal temperature gradient in the revised manuscript. We find that using central difference method in approximating the basal temperature gradient can have high-frequency oscillations between basal melting and freezing even if the climate forcing is constant in time, which may be an artifact of discretization (Figure S1). Therefore, we decide to use the one-sided difference method, and we will revise the methods accordingly.

*L224-226 I think that you have swapped around your "above" and "below" in this sentence. Surely the age modelling based on orbital tuning of the gas record is for the oldest, and therefore the deepest, part of the ice core, and the matching with AICC2012 is for the younger and shallower part of the core.*

This sentence is correct, particularly for the DF ice core chronology (Kawamura et al. 2017, Materials and methods, section of Chronology and stacking).

*L247-248 If the simulated temperatures are colder, especially in the middle of the ice column, this suggests that the downward advection of surface cold is probably too large, indicating that the p value might not be optimal. It might be worth adding a sentence here outlining this.*

We agree that the different p value can account for the temperature profile, but the change in p value affects the age profile, too. In addition, the temperature profile can also be affected by the parameterization of heat conductivity (comments of reviewer #1). We will revise this paragraph by discussing the factors (including downward heat advection) affecting the temperature profiles.

In all simulations, the simulated temperature profiles were generally colder than observed temperature profiles especially in the middle of the ice columns (Fig. 4a). The generally colder temperature of the ice shift may be derived from several reasons. One is the pressure melting point of the ice. We use pressure melting point of the ice depending only on local pressure, but it also depends on the impurities and air content of the ice. Another one is the vertical velocity of the ice parameterized with p, because a larger vertical advection contributes to a colder temperature of the ice.

*L272-273 Your estimate of an annual layer thickness of 0.1mm (Figure 6b, dark blue line) is for a GHF of 52 mW/m^2. You state on lines 250-251 that there has been no melt for a GHF of 52 mW/m2, therefore the age will be greater than 1.5Ma. At a minimum, you need to delete "of 1.5 MA BP ice" on line 272 because you don't know the age in this case.*

Yes, the annual layer thickness of 0.1mm (Figure 6b, dark blue line) is for a GHF of 52, not 55. In the GHF of 52 mW m$^{-2}$, there's no melt and the basal age is ~2.0 Ma. In this case, the 1.5 Ma ice appears at ~100 meters above the bedrock, so it is possible to define the resolution of 1.5 Ma ice. If the 1.5 Ma ice does not exist due to basal melting, it is impossible to define the resolution of the 1.5 Ma ice. We will revise the last sentences to clarify this.

Furthermore, suppose there was no significant basal melting, (Fig. 6b dark blue lines), the annual layer thickness of 1.5 Ma BP ice is approximately 0.1 mm because 1.5 Ma ice appears about ~100 meters above the bedrock.

*L302-314 It is somewhat ambiguous as to what you mean by "different amplitude of temperature changes", especially given your comment on lines 308-309 "because mean temperature over the glacial cycles increased if we reduce a small temperature amplitude of glacial-interglacial cycles." Presumably, this means that you have kept the interglacial temperatures unchanged and increased the glacial temperatures to change the "amplitude of the changes". If this is the case you should state this somewhere in Section 3.4*

Exactly. We have kept the interglacial temperatures unchanged in the smallest amplitude case. We will revise the sentences:

"The results using DF conditions with different amplitude of temperature changes but the same GHF and p parameters (same as Sect. 3.2) are summarized in Fig. 11 in terms of temperature and basal melting rates. Here, we changed the alpha value in Equation 8 by the factor of 0 to 0.75 (1 is the control case). In the smaller amplitude experiment (alpha=0), the temperature is set to the interglacial level and does not change in time. Note that the SMB variation is the same in all sensitivity experiments."

*L317-326 You might also want to mention that the GHF may vary over the spatial scale of the radar survey, (e.g. Carson et al 2013, doi:10.1144/jgs2013-030), especially given the sensitivity to GHF that you mention on line 276*

Thanks a lot for your suggestion. We will refer to the article(s) to state spatial variations in GHF based on radar surveys.

"Later in the article, we investigate the possibility of old ice in the DF region using different parameters of ice thickness and GHF, because glaciological surveys suggested there are spatial variations in these parameters (refereces)."

*L348-349 is the impact of the spatial distribution of SMB minor because 1) the sensitivity to SMB is low and/or 2) the spatial variability of SMB is low?*

It's mainly because the spatial variability of SMB is low. We clarify this:

"The results suggest that the spatial distribution of SMB (~20% for DF area) has a minor impact on the basal temperature compared with that of the ice thickness."

*L390 For the radar transect between DF and NDF, while the old ice occurs "where the ice is thin", this is at the expense of the age resolution. It would be good to add some words to point that out.*

We will add one sentence after this by referring Figure 13.

"We note that the thin ice is less good regarding the resolution of the old ice (Figure 13)."

*L466-470 The model-data discrepancy at 14-18 km from DF corresponds with a relatively cold ice-bedrock interface (Figure 15). This suggests that perhaps the estimated GHF of 55 mW/m^2 is too low locally, leading to cold ice with little/no basal melt and therefore vertical velocities that are too low. This is consistent with the model estimating ages that are too shallow. Such fine spatial scale GHF variations have been noted elsewhere in Antarctic, (see comment above for lines 317-326).*

Yes, a spatial distribution in GHF (as suggested above in L317-326) can be a source of model-data discrepancy, as well as others.

This model–data discrepancy indicates that the effects of vertical or horizontal advection (Huybrechts et al., 2007; Sutter et al., 2021), spatial distribution in geothermal heat flux or ice thickness changes over glacial cycles (Saito and 2020) may have contributed to this difference.

*L485-487 See comment above for L272-273*

Same as L272-273, if 1.5 Ma ice appears above the bedrock, it is possible to define the resolution of 1.5 Ma ice. We will revise the sentences to clarify that this sentences discuss Figures 13 and 14.

If the GHF is small enough to keep the basal temperature below the melting point, it is expected that ~1.5 Ma could be present. Based on Figs. 13 and 14, if such old ice exists, the simulated annual layer thickness of ~1.5 Ma BP ice is approximately 0.05 to 0.1 mm, corresponding to 10 to 20 ka m$^{-1}$.

Specific edits
===============
*L2 "around" -> "near"*

We are going to change the title as suggested.

*L29-30 This sentence could do with a reference, perhaps something like Shakun et al 2015, doi 10.1016/j.epsl.2015.05.042*

We will add references.

*L41 "critically scientific challenges" -> "critical scientific challenge"*

We will change the phrases as suggested.

*L59 "in the south" -> "to the south"*

"South" refers to specific areas rather than direction from DF site. We have changed the phrases to clarify this.

Subglacial mountains were detected in the area south of DF

*L63 it is unusual to talk about an "areal extent", i.e. an area and then give its size in units of length ("50km") rather than area.*

*L63 "NDF" has not be defined*

We will revise the phrase. And NDF is the name of the site. We will clarify this:

ice sheets over a distance of ~ 50 km, covering DF and NDF site, which locates south of DF (77.8° S, 39.05° E)

*L78 "Horizontal velocity" -> "Horizontal surface velocity"*

*L81 "experiments" -> "simulations"*

*L95-96 "convey the information of surface temperature" -> "advect and diffuse the surface temperature"*

We will change the phrases as suggested.

*L124-124 "zeta=s/H" -> "zeta=z/H"*

*L131 "ablation" -> "basal melt"*

Thank you for correcting.

*L138 delete "induce"*

We will change the phrases as suggested.

*L145-146 define "T" from equation 4*

We will define T in L145. .

*L159 "335,000 J kg^-1" -> "335 kJ kg^-1"*

*L242 Even though the section heading mentions "DF" it would be worth making it clear in the opening sentence. Suggest changing "temperature profiles" -> "DF temperature profiles"*

*L261 for clarity, suggest changing "reconstructed profiles" to "ice core based reconstructed profiles"*

*L268 suggest either deleting "as an indicator of old ice" or changing "as an indicator of old ice" -> "as an indicator of sufficient resolution for dating ice based on chemical and isotopic methods"*

      We will change the phrases as suggested, thank you.

*L289 "Table 2" -> "Table 1"*

      Thank you a lot for correcting this.

*L330 the results in section 3 included varying GHF, so therefore you need to delete "other"*

*L382-383 change "using seven colored lines" -> "for seven selected ages"*

      We will change the phrases as suggested.

*Figure 2 caption : "Equation [1]" should be "Equation [3]"*

      Thank you for correcting this.

*Figure 15 caption : need to include what "p" and GHF values are used for this experiment. Presumably p=3 and GHF=55 mW/m^2*

      Yes, p=3 and GHF=55 are used. We will add information on the experimental design in figure caption, thanks a lot.
* * *
REVIEWER #3

*Obase et al. present results for a 1D ice and heat flow model. The goal is to inform site selection for a new core site near Dome Fuji, targeting ice older than the ~700 ka limit of the previous core. The goals of the paper are to: 1) identify parameter combinations that approximately match the Dome Fuji depth-age and borehole temperature relationships and thus can be used for predicting depth-age relationships in the vicinity; 2) identify the primary constraints on the basal ages, which they determine is ice thickness; and 3) apply the model to the radar line that stretches from the previous ice*

*core site to a potential new site, North Dome Fuji.*

Thank you for your careful reading and giving us fruitful comments. We address your concerning comments below.

*I am providing only a brief review because I am concerned about the treatment of the basal thermal state in the model. In Figure 5, a change in the geothermal flux of 5 mW m-2 (from 55 to 60 mW m-2) yields a change in the average melt rate of ~2.5 mm/yr (from my eyeballing of the averages). This is too large. It should be about 0.5 mm/yr since 1 mW m-2 can melt approximately 0.1 mm/yr of ice. The caluclation is below:*

*the melt rate (M) equals the geothermal flux (G) divided by the latent heat (L) and the density of ice (ρ)*

$$M = G / L / \rho = 0.001 \ (W/m2) / 334000 \ (J/kg) / 917 \ (kg/m3)$$

*So I'm confused why the values in Figure 5 change so much for the modest increase in geothermal flux. I checked this with a model run of my own transient 1D ice and heat flow model with forcings for EDC based on AICC2012. The attached figure shows that modeled melt rate agrees with the calculation above – each 1 mW m-2 of excess geothermal flux causes approximately 0.1 mm/yr of basal melting.*

*I wonder if the Obase model has a problem with the basal boundary. It sounds like the temperature gradient is being set directly as the ice-rock boundary, instead of in the bedrock well below.*

*Unfortunately, the basal melt rate is the controlling factor on the depth-age, such that an error would affect the entire manuscript. I am not sure, but it looks like this problem is also affecting the depth-age relationship in Figure 6.*

*I initially wondering if there was some nonlinearity model that would amplify the basal melt rate in response to a change in geothermal flux. The basal melt rate affects the vertical velocity. But this has the impact of steepening the basal temperature gradient, allowing more of the heat to be conducted away rather than used to melt basal ice. So that works in the opposite direction. And the model run I performed suggests that there is not a significant non-linearity.*

Thanks a lot for the comments on model results. We found that this nonlinearity in basal melting comes from the central difference method in estimating basal temperature gradient by extrapolating ice temperature below bedrock. The method of discretization in basal temperature gradient is also commented by reviewer #2 (L161).

We have analyzed the heat budget in basal melting with the constant climate forcing (temperature and SMB are constant in time) for DF configuration. We compare results with calculating basal temperature gradient by one-sided discretization, and

central difference method used in the originally submitted article (Figure S3). In this idealized setting, GTH of ~53 mW/m2 is the threshold of basal melting (Figure S3a). And the basal melting rapidly increases when it starts melting by 2.5 mm $a^{-1}$ in 5 mW $m^{-2}$ if the central difference method is used (originally submitted manuscript). This behavior comes from a shift in the basal temperature gradient, as it significantly reduces above melting (Figure S3b-c). We also find the central difference method can have high-frequency oscillations between basal melting and freezing even if the climate forcing is constant in time, which may be an artifact from central difference method by extrapolating ice temperature into bedrock.

If we use one-sided difference method in estimating basal temperature gradient, the result of this case is similar as reviewer #3's results, in both of constant climate forcing (Figure S3) and DF case with realistic paleoclimate forcing. An excessive 5 mW $m^{-2}$ have a basal melting of 1 mm $a^{-1}$ according to the Revised Figure 5. The temperature gradient at the ice-bed interface plays a role in some nonlinearity in basal melting, but is less significant than the central difference method.

The revised manuscript will use a one-sided difference discretization method at the ice-bed interfaces. This change generally has smaller basal melting rates in same conditions, and affects the calibrated geothermal heat flux at DF conditions

[Figure]

Figure S3: (a) Simulated basal melting under constant climate forcing with two different methods in approximating basal temperature gradient. The red circles indicate results with central difference method in the original manuscript, and the black circles indicate results with one-sided difference method, which will be used in the revised manuscript. (b): Basal temperature gradient in the two different methods, (c) Temperature at ice-bed interface grids and one above and below ice-bed interfaces.

*The manuscript addresses an interesting problem of calculating the temporal variations in the basal melt rate and the impact on the depth-age relationship. However, I think the authors need to provide further support that they are calculating the basal melt rate accurately before the remainder of the manuscript is evaluated.*

Thanks a lot again for your careful reading and comments.

References:

Huybrechts and Payne (1996): The EISMINT benchmarks for testing ice-sheet models, Annals of Glaciology. doi:10.3189/S0260305500013197

Saito, F., and Abe-Ouchi, A. (2010) : Modelled response of the volume and thickness of the Antarctic ice sheet to the advance of the grounded area, Ann. of Glaciol., 51, 41-48, doi: 10.3189/172756410791392808, 2

---

## Author Response (AR1)

We thank all three reviewers who provided precise and valuable feedback on our manuscript. Our complete response is in this document. The reviewer's comments are quoted in italic, and our answers follow. The revised sentences of the manuscript are indicated in red text. We will be happy to submit a revised manuscript that reflects these changes.

In the revision of the manuscript, we have applied two major changes in the model experiments regarding the comments by reviewers. We have one minor change in the title of the manuscript ("near Dome Fuji", instead of "around Dome Fuji")

First, we have changed the numerical method of the ice-flow model in estimating basal temperature gradient in calculating basal melting rates, as suggested by reviewers #2 and #3. The revised manuscript will use a one-sided difference discretization method at the ice-bed interfaces in estimating basal temperature gradient instead of the central difference method used in the original manuscript.

Second, we have included the past ice thickness differences associated with the glacial-interglacial cycle in the transient simulations, as suggested by reviewer #1. The past ice thickness changes are from the output of 3-dimensional Antarctic ice sheet model simulations (Figure 3c). These two changes led to a lower basal melting rate in the same conditions compared to the methods used in the original submission. Because of the smaller basal melting rate, the calibrated geothermal heat flux at DF and EDC conditions is 60 mW m$^{-2}$, which is larger than in the original submission by ~5 mW m$^{-2}$ (Figure 6). This modification in the model impacts quantitative results regarding the possibility of old ice at given conditions, as summarized in a sensitivity experiment with different ice thicknesses (Figure 12 and 13).

We have revisited all experiments used in the article and changed all results figures accordingly (Figures 4-15). We have one minor change in the method of showing the resolution of age (annual layer thickness). The submitted article used the central difference of the simulated vertical age profiles to calculate age resolution. As one strength of the RCIP scheme is solving the spatial derivatives of the age simultaneously, the revised manuscript will use the simulated vertical derivatives of the age as the resolution of age. Although the difference from changing the method has a slight change in the result figures, we think showing the output from the RCIP scheme will be better.

Response to Frédéric Parrenin (REVIEWER #1)

*This manuscript presents simulations of a 1D age and temperature model, mainly for the Dome Fuji ice core and region, but also for the EDC ice core. The main aim of the manuscript, as the title reads, is to infer potential old ice drilling sites in the Dome Fuji region. The manuscript is generally of excellent quality. It is precise and reads well. However, I have a few suggestions for the authors which could further improve the relevance of the manuscript. I let the authors decide if they want to include these suggestions in their simulations, or simply discuss them in the discussion and outlook sections.*

Thank you for careful reading and giving us fruitful comments. We decide to adopt suggestions by conducting additional experiments and adding figures.

*Main comments:*

*- The model is interesting since it is a transient model, while other models used for the same purpose were steady (or pseudo-steady). However, the authors do not use the full power of this transient aspect of the model, since they fixed the ice thickness. As the authors wrote, the ice thickness is a primary parameter controlling the basal melting/temperature and therefore basal ice age. Therefore, a glacial-interglacial ice thickness change of 200 m can have an important impact on the simulations.*

Thanks a lot for suggestion. In the revised manuscript, we use a result of transient simulation obtained by a 3-D ice sheet model to simulate past ice thickness history. We use a 3-D ice sheet model IcIES, which computes dynamics and thermodynamics of ice sheets using the shallow-ice approximation. The experimental design is the same as Saito and Abe-Ouchi (2004; 2010) with some changes. We extracted the history of changes in the ice thickness changes at DF and EDC, which show that the ice thickness were reduced by ~200 meters during glacial periods (Fig. 3c). We conducted one set of sensitivity experiments with the absence of past ice thickness changes, which leads to a different past basal melting rates (Figure 11).

*- The authors find a shift between observed and simulated temperature profile near the bed.*
*They reckon that this is due to polythermal ice, but there is another explanation. Indeed, the pressure melting point is not so well known. Apart from pressure, it also depend on the impurities and air content of the ice. Catherine Ritz discussed that in a thesis 30 years ago, and this discussion is still relevant I think.*

We agree with this. We use pressure-melting temperature depends only on local pressure. Meanwhile, several studies use pressure-melting points with the function of pressure and air content. We have added sentences in the first paragraph of subsection 3.2. Furthermore, we have added sentences in subsection 3.3 (EDC) because pressure melting point dependency on air content of the ice is frequently used in EDC.

(L288-295, subsection 3.2): In all simulations, the simulated temperature profiles were generally colder than observed temperature profiles, especially in the middle of the ice columns (Fig. 4a). The generally colder temperature of the ice may have several explanations. One is related to the pressure melting point of the ice. We used a pressure melting point of ice that depended only on local pressure, but there is also a dependence on the impurities and air content of the ice (e.g., Parrenin et al., 2017; Passalacqua et al., 2017). A second explanation is related to the uncertainty in vertical velocity of the ice parameterized with p because a larger vertical advection contributes to a colder ice temperature.

(L331-335, subsection 3.3): The model generally resulted in colder temperatures compared with observations, similar to that found at DF (Fig. 7). We note that the pressure melting point of the ice depended only on local pressure in Fig. 7, but several studies have considered the pressure melting point of the ice as a function of the pressure and air content of the ice, which has shown that the basal temperature is at the pressure melting point (Buizert et al., 2021).

*- The 1D simulations for EDC are not discussed as much as the simulations for DF.*
*I understand the authors have deeper interests for Dome Fuji, but I think it could make the manuscript more valuable if the EDC case is discussed more. For example, I would have been interested by a graph showing the basal melting variations at EDC with time.*

Thank you for suggestion. We have added time-series of basal melting in EDC case in Figure 5b.

*- On the contrary, I did not find the simulation along the DF-NDF transect so interesting. To make it really interesting, it would have been necessary to invert the parameters (in particular accu and GHF) to fit the observed isochrones. There is no reason to assume accu varies linearly and GHF is constant.*

While one motivation for applying the 1-D model to the DF-NDF transect is examining the drill site, the role of the DF-NDF transect experiment in the present study is an exercise experiment under an idealized setting. We have clarified this in the first sentence of the chapter:

L425-426 (section 5.1): In this section, we apply the 1-D model to interpret the internal layers of the ice near DF under idealized boundary conditions.

*- The thermal parameters of the ice (conductivity, heat capacity) are not so well known. There are several parametrizations. Conductivity also depend on the fabric, which makes it even more challenging to estimate them. I think a discussion on these different parametrizations would have been valuable.*

We will write out the parameterizations of conductivity and heat capacity of the ice used in this study. (L157-158).

And we have added a discussion on the parameterizations of physical properties of the ice in the discussion:

L498-502: There is also uncertainty in the parameterization of the conductivity and heat capacity of the ice. We use these parameters as a function of temperature, but they can depend on the fabric of the ice, which makes it challenging to estimate them. Hence, these physical parameters can be a source of uncertainty in estimating GHF, and can be a source of difference from other studies.

*- Catherine Ritz showed a long time ago that it is best to simulate the temperature variations in the bedrock. Indeed, temperature waves propagates in the upper continental crust and the geothermal flux at the ice-bedrock interface cannot be assumed constant with time.*

The model in the present study forecasts temperature in the bedrock. We have clarified this in the model description:

L179-182: The model in the present study forecasts temperature in the bedrock, and thus the GHF at the ice–bedrock interface has temporal variations. The bedrock is 3000 m thick divided vertically into 17 equal layers; constant physical parameters are used for the bedrock (density = 2700.0 kg m−3, heat capacity = 1000.0 J kg−1K−1, and heat conductivity = 3.0 W m−1K−1).

*- It would have been interesting to make a Monte-Carlo simulation for DF and EDC to see which sets of parameters are acceptable. Here, the parameters are changed one after the other but there are probably covariances.*

We agree that Monte-Carlo simulation has an advantage in estimating a good set of parameters for specific sites at DF and EDC. Nevertheless, we would like to keep systematic sensitivity experiments in the present article rather than precise tuning with the previous DF ice core. This is because this study focuses on the range of glaciological

parameters around DF (ice thickness 2000-3200m, different SMB and GTH), which can differ from conditions from the previous DF ice core drilling site.

*Minor comments:*
*- l. 77-80: What is important for applying a 1D ice flow model is not the value of the horizontal velocity, but how the ice flow parameters (e.g., ice thickness) varies upstream. For example, an ice flow line can be 100 km long with a surface velocity of 1 m/yr, and a 1D model could still be appropriate if everything is constant upstream.*

We agree with this. We have added that the small spatial variations in horizontal ice velocity are also a factor in applying 1D models.

L80-82: Horizontal surface velocity in the vicinity of DF and NDF is < 2 m a–1, and it has minor spatial variations, evidenced by satellite-based measurements (Rignot et al., 2011, 2017; Mouginot et al., 2012).

*- l. 97-98: Parrenin et al. (2017) did not exactly assume that basal melting was constant. They used the pseudo-steady assumtion, which states that temporal variations in basal melting are the same than temporal variations of surface accumulation rate.*

We have corrected this:

L104-105: Parrenin et al. (2017) assumed that the temporal variations in basal melting rates are the same as accumulation rates.

*- eq. (2) and l. 130-131: I think there is an inconsistency here. As eq. (2) is written, a positive value of Mb means ice refreezing, not melting.*

Yes, negative indicates ice melt. We have revised this.

*- l. 133-134 and eq. (3): This equation was first formulated in Parrenin and Hindmarsh (2007).*

We have changed the sentence:

L142-144: The normalized vertical velocity profile, ω, is given as a function of the normalized coordinate derived from Parrenin and Hindmarsh (2007), and Llibtoury (1979):

*- l. 145-146: Regarding the neglecting of heat production, I think it could justified by the small ice deformation near a dome (very low horizontal shear which is the dominant factor elsewhere).*

We agree with this. We have changed the sentences:

L158-160: The strain heating term is neglected in the present study, given that deformation of the ice would be minor near Antarctic domes because of very low horizontal shear.

*- l. 146-148: There are different parametrizations of ice conductivity and thermal capacity (see comment above). These are not discussed here, but I reckon they can have an important effect.*

We have addressed the uncertainty in parameterizing in the second paragraph of the discussion when discussing uncertainties in estimating geothermal heat flux. Please see our reply below (comments on *L424-425*)

*- l. 148: Is it not 917 kg/m^3 the standard value for ice density? (note the wrong unit in the manuscript).*

We use 910 kg m$^{-3}$ as the standard value for the density of ice sheet. This value is one frequently used value in 3-dimensional ice sheet models (e.g., Huybrechts and Payne 1996, EISMINT ice sheet model inter-comparison project).

*- eq. (5) and (6) assumes a constant geothermal heat flux, which is not the case since heat waves propagate in the upper continental crust (see comment above).*

We have clarified that the geothermal heat flux at the ice-bedrock interface can have temporal evolutions.

L179-180: The model in the present study forecasts temperature in the bedrock, and thus the GHF at the ice–bedrock interface has temporal variations.

*- l. 164-166: I don't understand this sentence here. The formulation of the model does not allow for polythermal ice, so there is no reason to decrease the vertical resolution.*

We have changed the sentences for clarification:

L193-198: We have tested the sensitivity to the vertical resolution of temperature calculation and found that using fine vertical resolution leads to the formation of a temperature inversion layer in the bottom of the ice, which can be a significant error in estimating basal temperature gradient and basal melting. Therefore, we set the number of vertical layers of the model for thermodynamics as 100 (each approximately 30 m thick) to prevent the representation of temperature inversion layers.

*- l. 203-205: Could you please write the equation relating SAT and accu? Is it the saturation vapor pressure relationship?*

Yes, the equation relating SAT and precipitation is from the saturation vapor pressure using temperature above the atmospheric inversion layer. We have added sentences and equations (equations 11).

L224-226: We used past SMB as a function of temperature anomaly compared with the present day following Huybrechts and Oerlemans (1990), which is based on saturation vapor pressure:

*- l. 216: I find it a shame that the ice thickness is fixed despite the model being transient (see comment above).*

Thanks for suggestion. The revised manuscript use the ice thickness tendency term in the standard experiment.

*- l. 228-230: It would have been possible to initialize the age and temperature profile with steady profile, instead of constant values, for a faster convergence.*

We agree this. Meanwhile, assuming constant age and temperature is valid as the realistic glacial cycle forcing prevails over the entire ice column within approximately 100 ka. We have clarified this:

L262-264: These simplified initial conditions generated unrealistic temperature fields in the early stage of the simulation, but realistic glacial cycle forcing prevailed over the entire ice column within approximately 100 ka.

*- l. 247-248: The obs-model temperature shift near the bed is probably due to the formulation of the pressure melting point (see comment above).*

We have addressed this above. (L288-295 and L331-335)

- l. 261-263: For sure! Without basal melting, the age is infinite at the base.

Exactly. Meanwhile, the age of ice cannot exceed 2 Ma BP in this forward simulations initialized with the age of 0, we have clarified this:

L425-427: Note that the age of 2 Ma BP is the limit of the experiments, and the results indicate that the old ice exists 50 m above the bedrock if the ice thickness is thicker than ~2100 m.

*- l. 296: Parrenin et al. (2017) also estimated the GHF at EDC (Figure 5c), but the value (~60 mW/m^2) is far higher than what you obtained here.*

Yes, in the original submission the calibrated geothermal heat flux from at EDC (51 mW m$^{-2}$) was significantly smaller than Parrenin et al. (2017). After receiving all

reviewer's comments, we found that the method of calculating basal temperature gradient at ice-bed interface and the inclusion of the evolving ice thickness contributes to smaller basal melting. We have revisited the EDC experiments, and we found that the geothermal heat flux at EDC would be 56 mW m$^{-2}$ (Figure 8), which is now closer to Parrenin et al. (2017)'s values. There's still difference of ~4 mW m$^{-2}$ from Parrenin et al. (2017), which can be attributed to the difference in the history of basal melting, applying past climate of DF to EDC. We have addressed this in the manuscript.

L340-343: The estimated GHF at EDC is smaller than that given by Parrenin et al. (2017), who estimated it to be 60 mW m−2. This difference can be attributed to the difference in the history of basal melting, or the application of past climate history derived from DF to EDC.

*- l. 299: I find this paragraph a bit short (see comment above).*

We have clarified the logic of the sentences.

L343-344: The results from the application of our model to EDC suggest that it may be applicable to different glaciological conditions, particularly different ice thicknesses and SMBs.

*- l. 317-326: It would have been interesting to make a Monte-Carlo simulation to see which sets of parameters are acceptable (see comment above).*

We have addressed this above.

*- l. 364-379: This is a very simplified transect simulation (see comment above).*

We have addressed this above.

*- l. 424-425: Instead of using polythermal ice, use a different parametrization of pressure melting point.*

We have added sentences (same as comment above).

L498-502: There is also uncertainty in the parameterization of the conductivity and heat capacity of the ice. We use these parameters as a function of temperature, but they can depend on the fabric of the ice, which makes it challenging to estimate them. Hence, these physical parameters can be a source of uncertainty in estimating GHF, and can be a source of difference from other studies.

*- l. 460-462: Of course, Lilien et al. find different results since they simulated BELDC and not EDC, with a very different ice thickness.*

Exactly, the ice thicknesses at EDC and BELDC are quite different. We have clarified that the ice thickness is one critical factor when comparing the results with Lilien et al. (2021)'s.

L543-545: Therefore, the different ice thickness (3233 m for EDC) would be the most critical factor in the difference in the age resolution of 1.5 Ma ice when compared with Lilien et al. (2021), who used BELDC conditions (ice thickness of 2750 m).
* * *
REVIEWER #2

*"A one-dimensional temperature and age modeling study for selecting the drill site of the oldest ice core around Dome Fuji, Antarctica" by Obase et al. details experiments utilising one dimensional age and temperature modeling of the Antarctic ice sheet. The validity of the model is demonstrated by comparisons with ice-core based age reconstructions and temperature measurements at both Dome Fuji and the EPICA Dome C core sites. Parameter sensitivity and selection studies for the Dome Fuji region are then conducted, and finally the optimised model applied to a ground-based radar survey in the region, and the simulated age horizons compared to isochrones from the radar survey. Overall, the paper is well written and easy to follow and is worthy of publication in the Cryosphere, after minor re visions as detailed below.*

Thank you for your careful reading and giving us fruitful comments. We address point-by-point replies to all of your comments below.

*Minor issues*
*============*

*L83-85 The logic here isn't quite correct. While a lower accumulation rate is necessary to increase the number of years in a given thickness of ice, a lower accumulation rate will also reduce the vertical advection of cold from the surface down into the interior of the ice sheet, therefore increasing the temperature of the ice. So accumulation rate plays a dual and potentially competing role, but in terms of basal melt rates, lower accumulation is not necessarily a good thing.*

We agree that a smaller accumulation rate contributes to a chance of basal melting by decreased vertical advection of ice. As this was discussed by Fischer et al. (2013), we have revised the sentences:

L86-90: Their key finding was that melting at the base reduces the likelihood of old ice, and a lower ice thickness than that at previous ice core sites is a required condition

to avoid basal melting. Furthermore, a lower accumulation rate generally contributes to increasing the age of the ice at a certain height from the bedrock but increases the chance of basal melting, owing to the reduced vertical advection of cold ice.

*L104-106 Parrenin et al 2017 (doi:10.5194/tc-11-2427-2017) applied a time varying rate factor to both the accumulation and melt rates in there 1-D modelling around EDC. This rate factor was based on variations from the EDC ice core for the last 800ka and was constant before 800ka.*

We have corrected the sentences referring to Parrenin et al. (2017):

L104-105: Parrenin et al. (2017) assumed that the temporal variations in basal melting rates are the same as accumulation rates.

We have revised the following paragraph:

L112-114: Despite the close link between the temperature and age of ice owing to basal melting, the coupled simulations of thermodynamics and age of ice were not represented under transient climate forcing in previous modeling studies of old ice.

*L139-140 Need to make it clear that "m" is Fischer et al's equivalent to "p". Suggest re-wording from "in the case of m=0.5 in their study" to "where their parameter m fulfils a similar role to p in this study, the case of m=0.5"*
*L141-142 m=0.5 is only smaller than p=3 for zeta<0.3. Suggest re-wording from "with a smaller vertical velocity, particularly near the base of the ice" -> "with a smaller vertical velocity in the lower approximately third of the ice" or "with a smaller vertical velocity near the base of the ice"*

We have revised the sentences:

L149-152: Compared with Fischer et al. (2013), who used a different formulation of the vertical velocity profile with an m parameter (similar role as p of this study) of m = 0.5 (Fig. 2 dashed lines), p = 3 from Equation (3) gives a different vertical temperature profile, with a smaller vertical velocity, particularly near the base of the ice.

*L161 Are you really calculating the temperature gradient at ice-bedrock interface using a central difference? If so you would need to be modelling the temperature down into the bedrock. If you are doing this, you should mention that the thermal domain extends down into the bedrock and give the boundary conditions at the bottom of the rock domain. If you are only modelling the thermal domain in the ice, then you must be using a one-sided difference discretization at the ice-bedrock interface.*

The original manuscript use a central difference (by extrapolating ice

temperature below bedrock). As indicated by the top of this response letter, we have decided to use one-sided difference discretization in estimating basal temperature gradient in the revised manuscript. We find that using central difference method in approximating the basal temperature gradient can have high-frequency oscillations between basal melting and freezing even if the climate forcing is constant in time, which may be an artifact of discretization (Figure S1). Therefore, we decide to use the one-sided difference method, and we have revised the model description accordingly:

L176-178: This model assumes basal melting only occurs at ice–bedrock interfaces, and the temperature gradient at the ice–bedrock interface is calculated using a one-sided difference discretization.

*L224-226 I think that you have swapped around your "above" and "below" in this sentence. Surely the age modelling based on orbital tuning of the gas record is for the oldest, and therefore the deepest, part of the ice core, and the matching with AICC2012 is for the younger and shallower part of the core.*

This sentence is correct, particularly for the DF ice core chronology (Kawamura et al. 2017, Materials and methods, section of Chronology and stacking).

*L247-248 If the simulated temperatures are colder, especially in the middle of the ice column, this suggests that the downward advection of surface cold is probably too large, indicating that the p value might not be optimal. It might be worth adding a sentence here outlining this.*

We agree that the different p value can account for the temperature profile, but the change in p value affects the age profile, too. In addition, the temperature profile can also be affected by the parameterization of heat conductivity (comments of reviewer #1). We will revise this paragraph by discussing the factors (including downward heat advection) affecting the temperature profiles.

L28-295: In all simulations, the simulated temperature profiles were generally colder than observed temperature profiles, especially in the middle of the ice columns (Fig. 4a). The generally colder temperature of the ice may have several explanations. One is related to the pressure melting point of the ice. We used a pressure melting point of ice that depended only on local pressure, but there is also a dependence on the impurities and air content of the ice (e.g., Parrenin et al., 2017; Passalacqua et al., 2017). A second explanation is related to the uncertainty in vertical velocity of the ice parameterized with p because a larger vertical advection contributes to a colder ice temperature.

*L272-273 Your estimate of an annual layer thickness of 0.1mm (Figure 6b, dark blue line) is for a GHF of 52 mW/m^2. You state on lines 250-251 that there has been no melt for a GHF of 52 mW/m2, therefore the age will be greater than 1.5Ma. At a minimum, you need to delete "of 1.5 MA BP ice" on line 272 because you don't know the age in this case.*

The experiment with small GHF has no melt therefore the basal age is ~2.0 Ma. In this case, the 1.5 Ma ice appears at ~100 meters above the bedrock, so it is possible to define the resolution of 1.5 Ma ice. If the 1.5 Ma ice does not exist due to basal melting, it is impossible to define the resolution of the 1.5 Ma ice. We have revised the sentences:

L319-321: Furthermore, in a scenario with no significant basal melting, the annual layer thickness of 1.5 Ma BP ice is approximately 0.1 mm because 1.5 Ma ice appears directly above the bedrock (Fig. 6b, dark blue lines).

*L302-314 It is somewhat ambiguous as to what you mean by "different amplitude of temperature changes", especially given your comment on lines 308-309 "because mean temperature over the glacial cycles increased if we reduce a small temperature amplitude of glacial-interglacial cycles." Presumably, this means that you have kept the interglacial temperatures unchanged and increased the glacial temperatures to change the "amplitude of the changes". If this is the case you should state this somewhere in Section 3.4*

Exactly, the SAT is kept as interglacial temperatures in the smallest amplitude case. We have revised the sentences:

L356-360: The results using DF conditions with different amplitudes of temperature change but constant GHF and p parameters (GHF = 60 mW m−2 and p = 3) are summarized in Fig. 10. Here, we changed the α-value in Equation 10 (1 is the control case). In the smallest amplitude experiment (α = 0), the temperature was set to the interglacial level and did not change in time. Note that the SMB variation was the same in all sensitivity experiments.

*L317-326 You might also want to mention that the GHF may vary over the spatial scale of the radar survey, (e.g. Carson et al 2013, doi:10.1144/jgs2013-030), especially given the sensitivity to GHF that you mention on line 276*

Thanks for suggestion. We have mentioned that GHF may have spatial distribution by referring surveys.

L385-387: Later in the article, we investigate the possibility of old ice in the DF region using different parameters of ice thickness and GHF because glaciological surveys have suggested that there are spatial variations in these parameters (e.g., Carson et al., 2013).

*L348-349 is the impact of the spatial distribution of SMB minor because 1) the sensitivity to SMB is low and/or 2) the spatial variability of SMB is low?*

It's mainly because the spatial variability of SMB is low. We have clarified this:

L417-419: These results are generally consistent with those of Fischer et al. (2013), and suggest that the spatial distribution of SMB (~20% for the DF area) has a minor impact on the basal temperature compared with that of the ice thickness.

*L390 For the radar transect between DF and NDF, while the old ice occurs "where the ice is thin", this is at the expense of the age resolution. It would be good to add some words to point that out.*

We have added one sentence:

L463-464: On the basis of the results shown in Fig. 13b, we note that thin ice gives a poorer age resolution for the old ice.

*L466-470 The model-data discrepancy at 14-18 km from DF corresponds with a relatively cold ice-bedrock interface (Figure 15). This suggests that perhaps the estimated GHF of 55 mW/m^2 is too low locally, leading to cold ice with little/no basal melt and therefore vertical velocities that are too low. This is consistent with the model estimating ages that are too shallow. Such fine spatial scale GHF variations have been noted elsewhere in Antarctic, (see comment above for lines 317-326).*

Yes, a spatial distribution in GHF can be a source of model-data discrepancy, we have revised the sentence:

L539-541: This model–data discrepancy indicates that the effects of vertical or horizontal advection (Huybrechts et al., 2007; Sutter et al., 2021), or spatial distribution of GHF may have contributed to this difference.

*L485-487 See comment above for L272-273*

We have improved the sentences:

L567-570: If the GHF is small enough to keep the basal temperature below the melting point, it is expected that ~1.5 Ma ice could be present. According to Figs 14 and 15, the simulated annual layer thickness of ~1.5 Ma ice is approximately 0.05 to 0.1 mm, which corresponds to 10 to 20 ka m−1.

Specific edits
===============

*L2 "around" -> "near"*

We have changed the title as suggested.

*L29-30 This sentence could do with a reference, perhaps something like Shakun et al 2015, doi 10.1016/j.epsl.2015.05.042*

We have added reference.

*L41 "critically scientific challenges" -> "critical scientific challenge"*

We have changed the phrases.

*L59 "in the south" -> "to the south"*

"South" refers to specific areas rather than direction from the DF site. We have changed the phrases to clarify this.

L60-61: where subglacial mountains were detected in the area south of DF F

*L63 it is unusual to talk about an "areal extent", i.e. an area and then give its size in units of length ("50km") rather than area.*

*L63 "NDF" has not be defined*

We have revised the phrase. And NDF is the name of the site. We will clarify this:

L64-66: ice sheets over a distance of ~ 50 km, covering the DF and NDF sites (the latter located at 77.8° S, 39.05° E, south of DF) (Rodrigez-Morales et al., 2020).

*L78 "Horizontal velocity" -> "Horizontal surface velocity"*

*L81 "experiments" -> "simulations"*

*L95-96 "convey the information of surface temperature" -> "advect and diffuse the surface temperature"*

We have changed the phrases.

*L124-124 "zeta=s/H" -> "zeta=z/H"*

*L131 "ablation" -> "basal melt"*

We have corrected them.

*L138 delete "induce"*

We have changed the phrases.

*L145-146 define "T" from equation 4*

We have defined T. (L156)

*L159 "335,000 J kg^-1" -> "335 kJ kg^-1"*

*L242 Even though the section heading mentions "DF" it would be worth making it clear in the opening sentence. Suggest changing "temperature profiles" -> "DF temperature profiles"*

*L261 for clarity, suggest changing "reconstructed profiles" to "ice core based reconstructed profiles"*

*L268 suggest either deleting "as an indicator of old ice" or changing "as an indicator of old ice" -> "as an indicator of sufficient resolution for dating ice based on chemical and isotopic methods"*

We have changed the phrases or units.

*L289 "Table 2" -> "Table 1"*

We have corrected it.

*L330 the results in section 3 included varying GHF, so therefore you need to delete "other"*

*L382-383 change "using seven colored lines" -> "for seven selected ages"*

We have changed the phrases.

*Figure 2 caption : "Equation [1]" should be "Equation [3]"*

We have corrected it.

*Figure 15 caption : need to include what "p" and GHF values are used for this experiment. Presumably p=3 and GHF=55 mW/m^2*

We have clarified experimental design of Fig.15 in figure caption.

Fig. 15 caption: A combination of p = 3 and GHF = 60 mW m−2 is adopted in these experiments.
* * *
REVIEWER #3

*Obase et al. present results for a 1D ice and heat flow model. The goal is to inform site selection for a new core site near Dome Fuji, targeting ice older than the ~700 ka limit of the previous core. The goals of the paper are to: 1) identify parameter combinations that approximately match the Dome Fuji depth-age and borehole temperature*

*relationships and thus can be used for predicting depth-age relationships in the vicinity; 2) identify the primary constraints on the basal ages, which they determine is ice thickness; and 3) apply the model to the radar line that stretches from the previous ice core site to a potential new site, North Dome Fuji.*

Thank you for your careful reading and giving us fruitful comments. We address your concerning comments below.

*I am providing only a brief review because I am concerned about the treatment of the basal thermal state in the model. In Figure 5, a change in the geothermal flux of 5 mW m-2 (from 55 to 60 mW m-2) yields a change in the average melt rate of ~2.5 mm/yr (from my eyeballing of the averages). This is too large. It should be about 0.5 mm/yr since 1 mW m-2 can melt approximately 0.1 mm/yr of ice. The caluclation is below:*
*the melt rate (M) equals the geothermal flux (G) divided by the latent heat (L) and the density of ice (ρ)*

$$M = G / L / \rho = 0.001 \ (W/m2) / 334000 \ (J/kg) / 917 \ (kg/m3)$$

*So I'm confused why the values in Figure 5 change so much for the modest increase in geothermal flux. I checked this with a model run of my own transient 1D ice and heat flow model with forcings for EDC based on AICC2012. The attached figure shows that modeled melt rate agrees with the calculation above – each 1 mW m-2 of excess geothermal flux causes approximately 0.1 mm/yr of basal melting.*
*I wonder if the Obase model has a problem with the basal boundary. It sounds like the temperature gradient is being set directly as the ice-rock boundary, instead of in the bedrock well below.*
*Unfortunately, the basal melt rate is the controlling factor on the depth-age, such that an error would affect the entire manuscript. I am not sure, but it looks like this problem is also affecting the depth-age relationship in Figure 6.*
*I initially wondering if there was some nonlinearity model that would amplify the basal melt rate in response to a change in geothermal flux. The basal melt rate affects the vertical velocity. But this has the impact of steepening the basal temperature gradient, allowing more of the heat to be conducted away rather than used to melt basal ice. So that works in the opposite direction. And the model run I performed suggests that there is not a significant non-linearity.*

Thanks a lot for the comments on model results. We found that this nonlinearity in basal melting comes from the central difference method in estimating basal temperature gradient by extrapolating ice temperature below bedrock. The method of discretization in basal temperature gradient is also commented by reviewer #2 (L161).

We have analyzed the heat budget in basal melting with the constant climate forcing (temperature and SMB are constant in time) for DF configuration. We compare results with calculating basal temperature gradient by one-sided discretization, and central difference method used in the originally submitted article (Figure S1). In this idealized setting, GTH of ~53 mW/m2 is the threshold of basal melting (Figure S1a). And the basal melting rapidly increases when it starts melting by 2.5 mm $a^{-1}$ in 5 mW $m^{-2}$ if the central difference method is used (originally submitted manuscript). This behavior comes from a shift in the basal temperature gradient, as it significantly reduces above melting (Figure S1b-c). We also find the central difference method can have high-frequency oscillations between basal melting and freezing even if the climate forcing is constant in time, which may be an artifact from central difference method by extrapolating ice temperature into bedrock.

If we use one-sided difference method in estimating basal temperature gradient, the result of this case is similar as reviewer #3's results, in both of constant climate forcing (Figure S1) and DF case with realistic paleoclimate forcing (Figure 5 of the revised manuscript). An excessive 5 mW $m^{-2}$ have a basal melting of 1 mm $a^{-1}$ according to the Figure 5. The temperature gradient at the ice-bed interface plays a role in some nonlinearity in basal melting, but is less significant than the central difference method. Therefore, we have used one-sided difference discretization method at the ice-bed interfaces in the revised manuscript and we have revisited all experiments.

In the manuscript, we have clarified that an increase in basal melting rate of approximately 1 mm a−1 corresponds to every 5 mW m−2 increase in GHF in DF case.

L305-307: A larger GHF (≥ 60 mW m−2) results in basal melting occurring most of the time, with an increase in basal melting rate of approximately 1 mm a−1 for every 5 mW m−2 increase in GHF.

[Figure]

Figure S1: (a) Simulated basal melting under constant climate forcing with two different methods in approximating basal temperature gradient. The red circles indicate results with central difference method in the original manuscript, and the black circles indicate results with one-sided difference method, which will be used in the revised manuscript. (b): Basal temperature gradient in the two different methods, (c) Temperature at ice-bed interface grids and one above and below ice-bed interfaces.

*The manuscript addresses an interesting problem of calculating the temporal variations in the basal melt rate and the impact on the depth-age relationship. However, I think the authors need to provide further support that they are calculating the basal melt rate accurately before the remainder of the manuscript is evaluated.*

Thanks a lot again for your careful reading and comments.

References:

Huybrechts and Payne (1996): The EISMINT benchmarks for testing ice-sheet models, Annals of Glaciology. doi:10.3189/S0260305500013197

Saito, F., and Abe-Ouchi, A. (2010) : Modelled response of the volume and thickness of the Antarctic ice sheet to the advance of the grounded area, Ann. of Glaciol., 51, 41-48, doi: 10.3189/172756410791392808, 2

---

## Author Response (AR2)

*thank you for your thorough revision of your manuscript. You rigorously considered the comments by the referees and revised your calculations accordingly, thus clarifying major issues. Given the quality of the present version, I am happy to accept your manuscript in principle for publication in The Cryosphere. Below you find a list of comments regarding the revised version I ask you to consider and address before I can finally accept your manuscript for publication. The ATC2 pdf with the comments is attached as well.*

*Page and line refer to the Author Track Changes version 2 (ATC2).*

We appreciate the thoughtful review and comments. The editor's comments are quoted in italic, and our answers follow. The revised sentences of the manuscript are indicated in red text.

*32: remove past*

done

*57: BEDMAP2 datasets (Fretwell et al., 2013): Also in Bedmap3? In is available as ESSD revision, so worth pointing out.*

Yes, as JARE datasets are used in Bedmap 3, we have referred to it in the manuscript.

*63: ground-based*

done

*64: please use horizons instead of layers when refering to radar; layers have a finite thickness. Moreover, radar images boundaries in the ice (horizons). Although the term layer is widely used, I suggest to use the more accurate term horizon (or internal reflection horizon).*

Thanks for your suggestions. We have decided to use "horizons" throughout the article, but we hope to keep the term "internal layer" in parallel at the first appearance. Glaciological studies frequently use layer terms to express strata within ice sheets or firn. Internal reflections can occur due to layered changes (e.g., acidic layers) of permittivity or conductivity of ice sheets. The radar images give boundaries (horizons) within the ice sheet, meantime, we also observe layered structures from radar images. As the term "internal layer" has been used in a previous study related to oldest ice (e.g., Tsutaki et al., 2022), we would like to use the term in parallel at the first appearance.

JARE (2017–2018 and 2018–2019 Antarctic summers) conducted ground-based radar surveys to investigate the internal reflection horizons (internal layers) of ice sheets over

a distance of ~ 5650 km (Tsutaki et al. 2022)

*78 age distributions of ice.*
*79 change "as" to "to estimate"*
*80: domes*
We have changed as suggested

97: Do you want to mention that reasonable resolution is also important to be able to analyse the ice core?
Yes. We have changed the sentences for clarification:
The reasonable resolution of ice core containing climate signals which can be analyzed with current methods is important. Particularly, Saito et al. (2020) presented a numerical scheme of ice advection calculation for an improved representation of annual layer thickness of the ice, and conducted numerical simulations using idealized glacial cycle forcings.

*105: used the pseudo steady-stage assumption, i.e.*
We have changed the sentence as suggested:
Fischer et al. (2013) used pseudo steady-state assumption, i.e., a constant climate forcing.

*135: "where s is the surface elevation" - s is not used anymore, please correct*
We have deleted definition of s, as it is unnecessary.

*152: an -> a*
*166: K−1. (colon)*
We have corrected them

*188: any references for these values?*
We have add the reference for the parameters (Parizek and Alley, 2004).

*235: I understand SMB as the akronym. It would be more consistent to use a letter as the actual surface mass balance rate variable*
We changed equation 11, using "a" to represent SMB.

*(e.g. ¥dot b, compare also Table 1) than using the informatic slang SMB (which one might use in a program) in an equation. Moreover, "SMB rate" would also be more consistent*

*with what you use for basal melting, where you use rate (see line 311 in ATC2).*
*SMB rate (in table 2 and manuscripts)*
*291: balance ->> balance rate (as you use mm/a)*
*292: balance rate*
*293: balance rate*
We have changed to "SMB rates" in Tables 1 and 2, and captions therein.

*256: "We used ... history; we used ..." - Wording a bit strange, please rephrase*
We have rephrased the sentence:
We used a result of transient simulation obtained by a 3-D ice sheet model IcIES, which computes dynamics and thermodynamics of ice sheets using the shallow-ice approximation to simulate past ice thickness history.

*472: change format to TC style: 17 December 2017*
done

*491: Please provide information on the traveltime-depth conversion of the radar data (e.g. wave speed, firn correction) shown in Fig. 15.*
This information is described in Tsutaki et al., (2022). We briefly provide it in this manuscript.
The ice thickness at the time of observation was converted from the two-way travel time from the surface to the ice-bed interface under the assumption of a propagation velocity, with calibration at DF ice coring site (Tsutaki et al., 2022).

*497: layer -> horizon*
*503: The reader cannot see the traced horizon, so cannot see the difference. Either provide an age on the left y-axis (from DF) or indicate this particular layer of 128 ka BP mentioned in the text also in Fig. 15.*
*For consistency please replace tracked by "traced" and layer by horizon (see later in ATC2).*
We have changed layer to horizon. We have revised Figure 15 and show rough estimate of the horizon corresponding ~128 ka.

*542: temperature -> air temperature*
done

*568: "should be complex": It is a bit unclear what you mean - should be considered more complex in the simulation rather than simple? In the previous sentence you already say that deformation is complex. Please try to clarify.*

We have revised the sentences for clarification:

According to analyses of the DF ice core (Azuma et al., 1999; Saruya et al., 2022) or 3-D ice sheet modeling (Seddik et al., 2011), deformation of the ice or flow regime towards the bottom of the ice is complex, suggesting parameterizing vertical velocities is difficult particularly near ice bottom. Improving velocity fields in ice sheet model would be an important issue for future studies.

*571: I do not agree that resolution is an indicator - either the ice is there or not. Resolution is a requirement to obtain an ice core which 1) still contains climate signals which were not destroyed by e.g. diffusion and 2) can be analysed with current methods. Easiest would be to remove "one indicator of old ice". If I misunderstand something here please clarify.*

We agree that the resolution itself is not an indicator of old ice. We have removed "one indicator of old ice" here to avoid confusion.

*573: 2750 should be 2765 (Lilien et al provide 2764 +/- 20 m, so 2750 is not justified as an approximate ice thickness imo).*
*573: I would also add: "2765 m, including a thickness of a basal unit of ~200 m and thus an effective ice thickness of 2565 m)". The free lower boundary is an important difference to your model approach.*
*574: Please remove "in contrast". You are at a different site (EDC), so you cannot claim to have contrasting results with BELDC.*
*579: Please add "... profile. In addition, Lilien et al. (2021) also allowed a finite thickness of a basal unit, which further reduced the effective ice thickness." As mentioned above, this is an important difference.*

We have revised the sentences using your phrases:

It is worth mentioning that the approach in ice thickness are different between Lilien et al. (2021) which used ice thickness of 2765 m, including a thickness of a basal unit of ~200 m and thus an effective ice thickness of 2565 m.

*580: Figs -> Fig.s*
*581: ice -> effective ice*
*582: in -> for*

*583: in -> of*
*585: ice -> effective ice*
*585: 2750 -> 2565*
Done

*589: horizons (a layer has a finite thickness, whereas a horizon is created by a reflection at the interface of two layers with different properties)*
*591: layers -> horizons*
*591: traced: please show this traced horizon in the figure*
We have changed sentences using the term horizon. We have changed Figure 15 to show the traced horizon.
~150 m above the isochrone horizons traced from DF

*593: distribution -> variation*
*601: reflection horizons*
We have changed as suggested.

*605: . -> :*
*609: Fig.s*
done

*635: information on glacial-interglacial times scales.*
We have added the phrase.

*643: altitude should be changed to height for consistency with other figures*
*Fig. 2: y-axes should read height instead of altitude for consistency with other figures*
We have changed them. (Altitude -> height)

*Fig. 4: in other figures you use Kelvin as a unit. I suggest to use also K here on the x-axis in order to avoid a confusion with the degree scale, which for instance can easily happen in Fig. 15 (e.g.-2°C could be interpreted as the actual temperature rather than -2° below the PMP). Same suggestion also for other figures (a suggestion, up to you to decide).*
We have changed $°C$ -> K in Figures 4, 7, 9-11, 15.